# The push–pull intercrop *Desmodium* does not repel, but intercepts and kills pests

**Anna L Erdei[1,2†], Aneth B David[1,3†], Eleni C Savvidou[1,4], Vaida Džemedžionaitė[1], Advaith Chakravarthy[1], Béla P Molnár[2], Teun Dekker[1]\***

[1]Department of Plant Protection Biology, Swedish University of Agricultural Sciences, Alnarp, Sweden; [2]Department of Chemical Ecology, HUN-REN Centre for Agricultural Research Plant Protection Institute, Budapest, Hungary; [3]Department of Molecular Biology and Biotechnology, University of Dar-es-Salaam (UDSM), Salaam, United Republic of Tanzania; [4]Department of Agriculture Crop Production and Rural Environment, University of Thessaly, Volos, Greece

**\*For correspondence:**
teun.dekker@slu.se

[†]These authors contributed equally to this work

**Competing interest:** The authors declare that no competing interests exist.

**Abstract** Over two decades ago, an intercropping strategy was developed that received critical acclaim for synergizing food security with ecosystem resilience in smallholder farming. The push–pull strategy reportedly suppresses lepidopteran pests in maize through a combination of a repellent intercrop (push), commonly *Desmodium* spp., and an attractive, border crop (pull). Key in the system is the intercrop's constitutive release of volatile terpenoids that repel herbivores. However, the earlier described volatile terpenoids were not detectable in the headspace of *Desmodium*, and only minimally upon herbivory. This was independent of soil type, microbiome composition, and whether collections were made in the laboratory or in the field. Furthermore, in oviposition choice tests in a wind tunnel, maize with or without an odor background of *Desmodium* was equally attractive for the invasive pest *Spodoptera frugiperda*. In search of an alternative mechanism, we found that neonate larvae strongly preferred *Desmodium* over maize. However, their development stagnated and no larva survived. In addition, older larvae were frequently seen impaled and immobilized by the dense network of silica-fortified, non-glandular trichomes. Thus, our data suggest that *Desmodium* may act through intercepting and decimating dispersing larval offspring rather than adult deterrence. As a hallmark of sustainable pest control, maize–*Desmodium* push–pull intercropping has inspired countless efforts to emulate stimulo-deterrent diversion in other cropping systems. However, detailed knowledge of the actual mechanisms is required to rationally improve the strategy, and translate the concept to other cropping systems.

## Editor's evaluation

This study addresses both commonly accepted and alternative hypotheses for the mechanism by which an intercrop supports pest control in push–pull agriculture, a promising and broadly recognized approach for sustainable intensification. The findings address a widely recognized gap in data on the mechanism underlying push–pull systems and thus can be important for work on pest control in agroecology as well as plant–herbivore interactions more generally. The support of claims is solid, combining observations of several different mechanistic aspects in an uncommonly broad range of relevant environments with clear reasoning regarding experimental design.

## Introduction

Since the dawn of agriculture, humanity has been in an arms race with insect pests. Traditionally, a set of integrated cultivation strategies tailored to local settings helped keeping pest insects at bay,

including associational resistance through varietal mixtures and intercropping (**Abate et al., 2000**; **Snyder et al., 2020**; **Wuest et al., 2021**). With the advent of agrochemicals, monocultures superseded traditional strategies. However, their profound externalities on ecosystem resilience and global climate (**Altieri, 2009**; **Shukla et al., 2019**) have resuscitated interest in more sustainable alternatives, frequently grafted on traditional strategies. Trending terms such as agroecology, and climate-smart, regenerative, or organic agriculture highlight the search for solutions that harmonize food production and pest control with ecological sustainability. Some innovative practices have been important sources of inspiration. Among these, the push–pull strategy in which maize is intercropped with the legume, *Desmodium*, is arguably the most well known (**Cook et al., 2007**).

The push–pull strategies aim to reduce the abundance of insect pests in crops through repelling the ovipositing herbivores from the crop, while simultaneously attracting the pest outside the field (**Miller and Cowles, 1990**). Using this 'stimulo-deterrent diversion' principle, a push–pull strategy was devised to combat lepidopteran pests in sub-Saharan smallholder maize farming (**Khan et al., 1997b**; **Khan et al., 2010**). Embroidering on the common practice of smallholder farmers to intercrop maize with e.g. edible pulses, the strategy uses the perennial fodder legume *Desmodium* as intercrop in maize plots. *Desmodium* reportedly constitutively releases large amounts of volatile monoterpenes and sesquiterpenes, such as (*E*)-4,8-dimethyl-1,3,7-nonatriene ((*E*)-DMNT), (*E*)-β-ocimene and cedrene, that repel (push) lepidopteran pests and attract natural enemies (pull) (**Hassanali et al., 2008**; **Khan et al., 1997a**; **Khan et al., 2000**). A 'trap crop' sown as border crop (another 'pull' component), typically Napier grass, complements the strategy, as it induces oviposition in *Lepidoptera*, but reduces larval survival compared to maize (**Khan et al., 1997a**; **Khan et al., 2000**; **Khan et al., 2016**). This cropping strategy thus suppresses infestations with various lepidopteran pests, including *Chilo partellus* and *Busseola fusca*, as well as *Spodoptera frugiperda*, a polyphagous invasive pest that is ravaging maize and vegetable production and threatens food security in sub-Saharan Africa (**Midega et al., 2018**; **Feldmann et al., 2019**). This intercropping strategy has found widespread adoption in East Africa (**Khan, 2011**; **Niassy et al., 2022**; **Nkurunziza, 2021**; **ICIPE, 2019**; **Government of Ruanda, 2011**; **Kenya National Assembly, 2019**). As a hallmark of sustainable pest control, it also serves as a tremendous source of inspiration for intervention strategies in other cropping systems.

The 'push' volatile terpenoids reported in previous studies (**Khan et al., 1997a**; **Khan et al., 2000**) are usually released in detectable amounts by plants after induction by herbivory. Although several plants do release them constitutively, such as *Melinis minutiflora* (**Kimani et al., 2000**), constitutive release of volatile terpenoids is not known from legumes. We sought to understand the role of soil-borne interactions, particularly the soil microbiome, in the constitutive release of these volatiles. This is of particular interest given that push–pull intercropping of maize and *Desmodium* causes substantial shifts in below-ground ecosystems, including increased soil microbe diversification, increased soil nitrogen and carbon, increased plant defense through plant–soil feedback, and suppression of parasitic weeds and pathogenic microbes (**Mutyambai et al., 2019**; **Mwakilili et al., 2021**). Indeed, soil and root–microbe interactions have been found to induce pathways that lead to release of volatile terpenoids (**Mutyambai et al., 2019**; **Malone et al., 2020**). We therefore verified if the 'constitutive' release of volatile terpenoids was, in fact, induced or enhanced by soil-borne interactions. The root–microbe interactions are of particular interest, given the intimate association of legumes with specific microbial groups e.g. rhizobia and mycorrhizae.

## Results and discussion

Different from the expectations, in the headspace of intact *Desmodium intortum*, which is by far the most commonly used intercrop in push–pull technology, the presence of the earlier described (**Hassanali et al., 2008**; **Khan et al., 1997a**; **Khan et al., 2000**) volatile terpenoids was not detectable (**Figure 1A, B**, **Figure 1—figure supplements 1 and 3**). This was independent of the soil in which *D. intortum* was grown, whether live soil (organic potting soil, organic clay Swedish soil, or African clay loam soil from *D. intortum* plots), autoclaved soil, or autoclaved soils inoculated with mycorrhiza or rhizobacteria (**Figure 2—figure supplements 1–4**) was used. Similar results were obtained with *D. uncinatum*, a species that has also frequently been used in push–pull cropping systems (**Figure 1—figure supplement 2**). In contrast, we did confirm that *M. minutiflora*, a poacean plant used previously as a push intercrop, releases a diverse blend of terpenoids regardless of herbivory ( **Figure 1—figure**

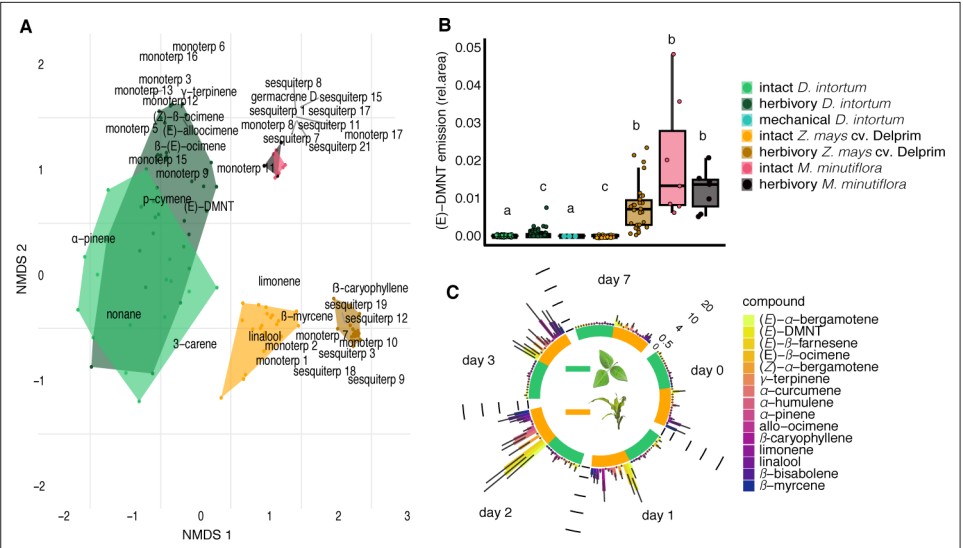

**Figure 1.** *Desmodium intortum* does not constitutively release terpene volatiles, and hardly following larval feeding. (**A**) Non-metric multidimensional scaling (NMDS) analysis of volatiles emitted by *D. intortum*, *Z. mays* cv. Delprim, and *M. minutiflora* plants, intact and 48 hr following *S. frugiperda* feeding (stress value = 0.138). (*E*)-4,8-Dimethyl-1,3,7-nonatriene ((*E*)-DMNT), (*Z*)-β-ocimene, (*E*)-β-ocimene, and (*E*)-alloocimene were not constitutively released, and only in low quantities in response to herbivory. Volatiles emitted by intact and herbivore-induced *D. intortum* (n=7 each, $F_{model}$ = 15.597, $R^2$ = 0.132, $p_{adj}$ = 0.021) and *Z. mays* plants (n=7 each, $F_{model}$ = 50.521, $R^2$ = 0.512, $p_{adj}$ = 0.021) were significantly different in PERMANOVA and pairwise comparison, but emissions from intact and herbivore-induced *M. minutiflora* plants (n=7, $F_{model}$ = 1.469, $R^2$ = 0.109, $p_{adj}$ = 1) were not. (**B**) (*E*)-DMNT emission before and 48 hr following herbivory (*n* = 8, errir bar represents ± standard error [SE]). The absolute peak areas were divided by the peak area of the internal standard and divided by the sum of monoterpenoids across all laboratory volatile collections for normalization. Treatments with different letters are statistically different (Kruskal–Wallis with Benjamini and Hochberg p value correction, $\chi^2$ = 57.315, p = 1.578 × $10^{-10}$). (**C**) Emission of volatile monoterpenoids and sesquiterpenoids from *D. intortum* and *Z. mays* before, during, and after *S. frugiperda* larval feeding (*n* = 5, error bars represent ± standard error [SE]). Peak areas of each terpenoid were divided by the area of the internal standard and divided by the sum of monoterpenoids or sesquiterpenoids across all laboratory volatile collections. Error bars show the standard error for relative volatile emission of each group. Day 0 – volatile emission before herbivory, Day 1 – 24 hr after herbivory, Day 2 – after 48 hr, and so on. Larvae were removed after 48 hr. *Figure 1—figure supplement 1*. Heatmap showing relative amounts of headspace volatile compounds emitted from intact, herbivore induced and mechanically damaged *Desmodium intortum*, *Zea mays* cv. Delprim, and *Melinis minutiflora* plants grown in a greenhouse. *Figure 1—figure supplement 2*. Volatile emission of *Desmodium uncinatum* and *Desmodium intortum* compared to *Melinis minutiflora* and *Zea mays* cv. Delprim. *Figure 1—figure supplement 3*. Ordination of volatile samples from intact, herbivore damaged, and mechanically damaged *Desmodium intortum*, *Zea mays* cv. Delprim, and *Melinis minutiflora* plants based on non-metric multidimensional scaling (NMDS).

The online version of this article includes the following figure supplement(s) for figure 1:

**Figure supplement 1.** Heatmap showing relative amounts of headspace volatile compounds emitted from intact, herbivore induced and mechanically damaged *Desmodium intortum*, *Zea mays* cv. Delprim and *Melinis minutiflora* plants grown in a greenhouse.

**Figure supplement 2.** Volatile emission of *Desmodium uncinatum* and *Desmodium intortum* compared to *Melinis minutiflora* and *Zea mays* cv. Delprim.

**Figure supplement 3.** Ordination of volatile samples from intact, herbivore damaged, and mechanically damaged *Desmodium intortum*, *Zea mays* cv. Delprim and *Melinis minutiflora* plants based on non-metric multidimensional scaling (NMDS).

---

*supplements 1–3*). Intact *Desmodium* plants thus did not release the earlier described repellent compounds in detectable quantities, independent of soil interactions.

Although the constitutive release of repellent volatile terpenoids is an important precondition for push–pull, inadvertent herbivory of *Desmodium* could result in a volatile emission similar to those reported in earlier studies for intact plants (*Khan et al., 2000*). In our experiments, only small amounts

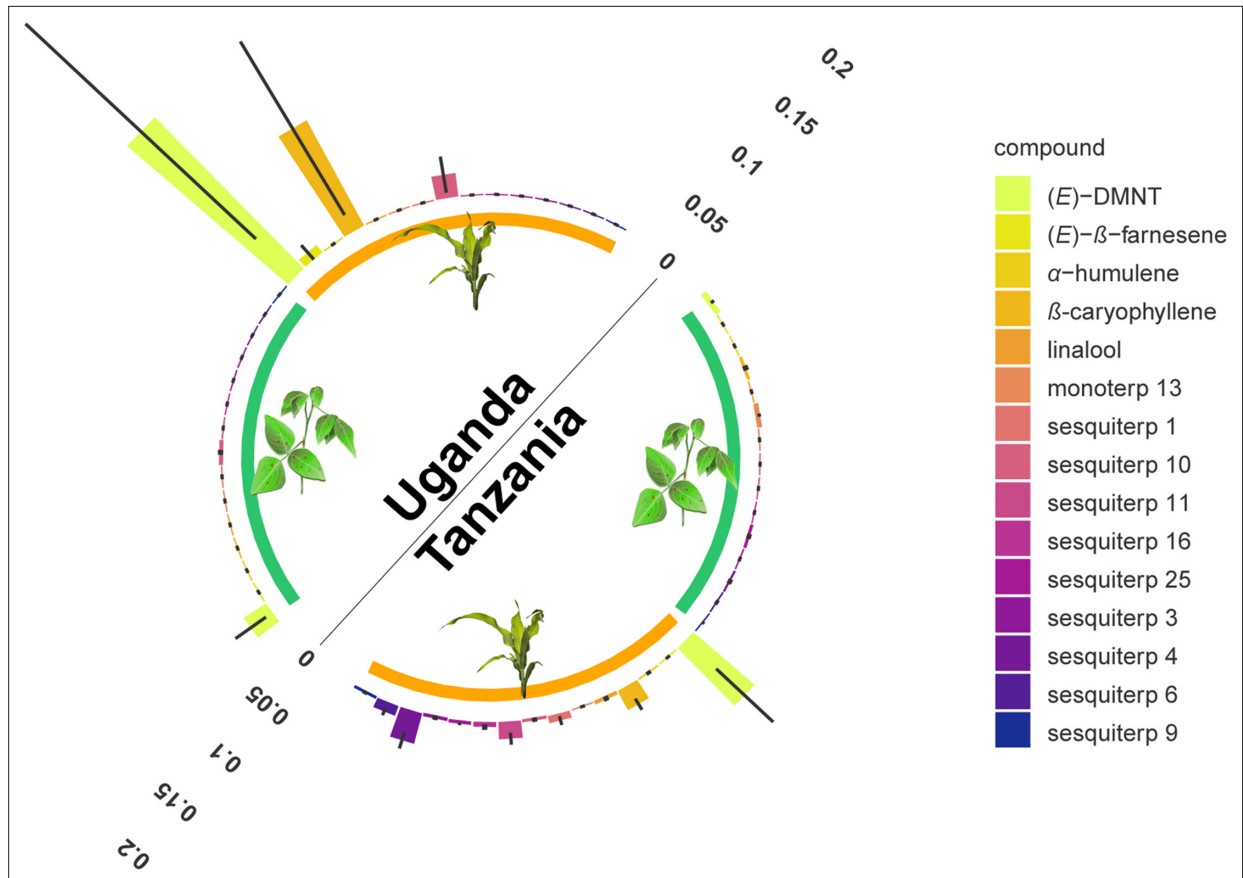

**Figure 2.** Monoterpenoid and sesquiterpenoid emission by *D. intortum* and *Zea mays* plants under field conditions at several locations in Tanzania and Uganda. The absolute peak area of each peak was divided by the sum of the area of monoterpenoid or sesquiterpenoid emission across all samples from the same location. Error bars represent ± standard error (SE) on the scale of the relative volatile emission (*Desmodium*: n = 17 and 20, maize = 5 and 15, for locations in Uganda and Tanzania, respectively). Minor terpenoid compounds were not identified to species level as this was not the focus of the study, and was further hindered by the vast diversity of compounds and the lack of synthetic standards. The volatile terpenoids were infrequently observed in the headspace of intact *D. intortum* plants regardless of soil microbial inoculation. (*Figure 2—figure supplement 1*) Volatile emission of field grown *Desmodium intortum* and *Zea mays* plants from two locations. (*Figure 2—figure supplement 2*) Volatile emission profile of intact and herbivore-damaged *Desmodium intortum* and *Zea mays* grown in soils with different microbial composition. (*Figure 2—figure supplement 3*) The infrequent observation of volatile terpenoids in the intact *Desmodium intortum* does not result from poor soil microbiota and insufficient nodulation. (*Figure 2—figure supplement 4*) The emission profile of *Desmodium intortum* and *Zea mays* cv. Delprim was not significantly altered by soil microbial treatments.

The online version of this article includes the following figure supplement(s) for figure 2:

**Figure supplement 1.** Volatile emission of field grown *Desmodium intortum* and *Zea mays* plants from two locations.

**Figure supplement 2.** Volatile emission profile of intact and herbivore-damaged *Desmodium intortum* and *Zea mays* grown in soils with different microbial composition ( n = 7).

**Figure supplement 3.** Terpenoids were rarely detected in the volatile samples of intact *Desmodium* plants and the volatile profile of plants grown in soil inoculated did not differ significantly.

**Figure supplement 4.** The emission profile of *Desmodium intortum* and *Zea mays* cv. Delprim was not significantly altered by soil microbial treatments.

of volatile terpenoids were detected in the headspace of *D. intortum* plants when fed upon by *S. frugiperda* larvae (*Figure 1A-C*; *Figure 2*). This is in contrast with maize, which, in line with previous studies (*Turlings et al., 1990*; *Degen et al., 2004*), was detected to release large amounts of volatile terpenoids in response to herbivory, with emission peaking between 24 and 48 hr following infestation, and declining over the course of 7 days (*Figure 1C*). Herbivory of *M. minutiflora* did not significantly boost the release of volatile terpenoids above the already high constitutive release (*Figure 1B*, *Figure 1—figure supplements 1 and 3*).

Arguably, greenhouse conditions are not representative of field conditions and additional, unknown factors in the field may cause the release of volatile terpenoids by *Desmodium*. We therefore analyzed 50 headspace samples from *D. intortum* from seven locations in Tanzania and Uganda. Also under field conditions, very few *D. intortum* plants released detectable amounts of terpenoids in the headspace. The few plants that did release terpenoids, did so in comparatively low amounts (*Figure 2*, *Figure 2—figure supplement 1*). This was most likely induced by herbivory, which upon further inspection was visible on these sampled plants. This confirms the greenhouse experiments and underlines that *Desmodium* does not constitutively release detectable amounts of volatile terpenoids, whereas following induction, the release is comparatively low. Although it cannot be excluded that other conditions or more substantial herbivore attack may induce higher release of volatile terpenoids, our experiments conducted under different growth conditions, and in different geographic regions show that this is likely rare, which would make it tenuous to be at the core of a generic strategy. In contrast, the headspace of field-sampled maize, most of which displayed some herbivore damage, did contain typical herbivore-induced volatile terpenoids (*Turlings et al., 1990*; *Degen et al., 2004*; *von Mérey et al., 2013*; *Figure 2*, *Figure 2—figure supplement 1*), with variations in volatile release likely reflecting differing levels of, and age since herbivore infestations, which could not be controlled in the field. The findings reported here are results of greenhouse experiments and field experiments from three geographical areas (Tanzanian highlands and lowlands, and Uganda) involving a variety of abiotic and biotic factors (including genetic background of *D. intortum* and potential herbivory). However, how these, and other environmental factors may have influenced volatile emission have not yet been investigated specifically and warrant further study.

Although volatile terpenoids were sparsely observed in the headspace of *Desmodium*, and seemed to be an unlikely cause of oviposition repellence, we tested the oviposition repellency in bioassays. In a modified wind tunnel, gravid *S. frugiperda* were given a choice between maize plants with either *D. intortum* or artificial plants in the background (*Figure 1D*, *Figure 3—figure supplement 1*). Adult females landed on either maize plant and the number of egg batches were not significantly different, underlining that odor from *D. intortum* that was placed upwind from the maize plants, did not elicit significant oviposition repellence in gravid *S. frugiperda* (*Figure 3B*). Contrary to our findings, *D. intortum* volatile emission appeared to be repellent for *S. frugiperda* in a recent study (*Sobhy et al., 2022*). However, the volatile profile of the *D. intortum* plants in the choice tests (*Sobhy et al., 2022*) was very different from intact plants we have studied and was reminiscent of the *Desmodium* plants under active herbivory infestation by larvae (*Figure 1—figure supplement 1*).

In order to explain the suppression of lepidopteran pests using *Desmodium* as intercrop, one needs to invoke a different mechanism than odor-based 'stimulo-deterrent diversion' or 'push–pull'. To investigate possible alternatives we scored *S. frugiperda* oviposition preference, larval feeding preference, and larval survival on maize and *Desmodium*. First, in two-choice tests *S. frugiperda* preferred to lay eggs on maize over *Desmodium*. Yet, the preference was not strong, as females also oviposited on *Desmodium*. In the field, one could perhaps expect a further shift toward *Desmodium*, particularly when maize is small and *Desmodium*, a perennial, well developed.

Though, irrespective of female oviposition choice, larvae of many lepidopteran species are known to disperse from the plant on which they hatched. Neonate, first instar larvae rapidly disperse to avoid sibling competition. Besides locomotion, they also passively disperse with wind through spinning silk threads allowing them to 'parachute' between plants (*Njuguna et al., 2021*; *Rojas et al., 2018*; *Sokame et al., 2020*). Later larval stages, which no longer disperse with wind, have been observed to actively disperse across the soil in search for new host plants (*Sokame et al., 2020*; *Berger, 1994*; *van Rensburg et al., 1988*). Given the dense, contiguous ground cover provided by *Desmodium* in the interrows, stochastically a large majority of dispersing larvae would end up on *Desmodium*, particularly when maize plants are small and *Desmodium*, a perennial, large. We asked whether these larvae would feed and survive on *Desmodium*. In feeding choice assays, the number of first instar *S. frugiperda* larvae on leaf discs and the area consumed was significantly higher for *Desmodium* compared to maize (*Figure 4*). However, in survival analyses, the development of those fed on *Desmodium* stagnated, with hardly any larva molting to the second instar, and none reaching pupation (*Figure 5*). Several sensory modalities including vision (*Han et al., 2024*), olfaction (*Castrejon et al., 2006*), taste (*Castrejon et al., 2006*; *Sun et al., 2022*), and tactile stimuli can potentially influence the consumption patterns observed in the larval choice assays. While further studies are needed

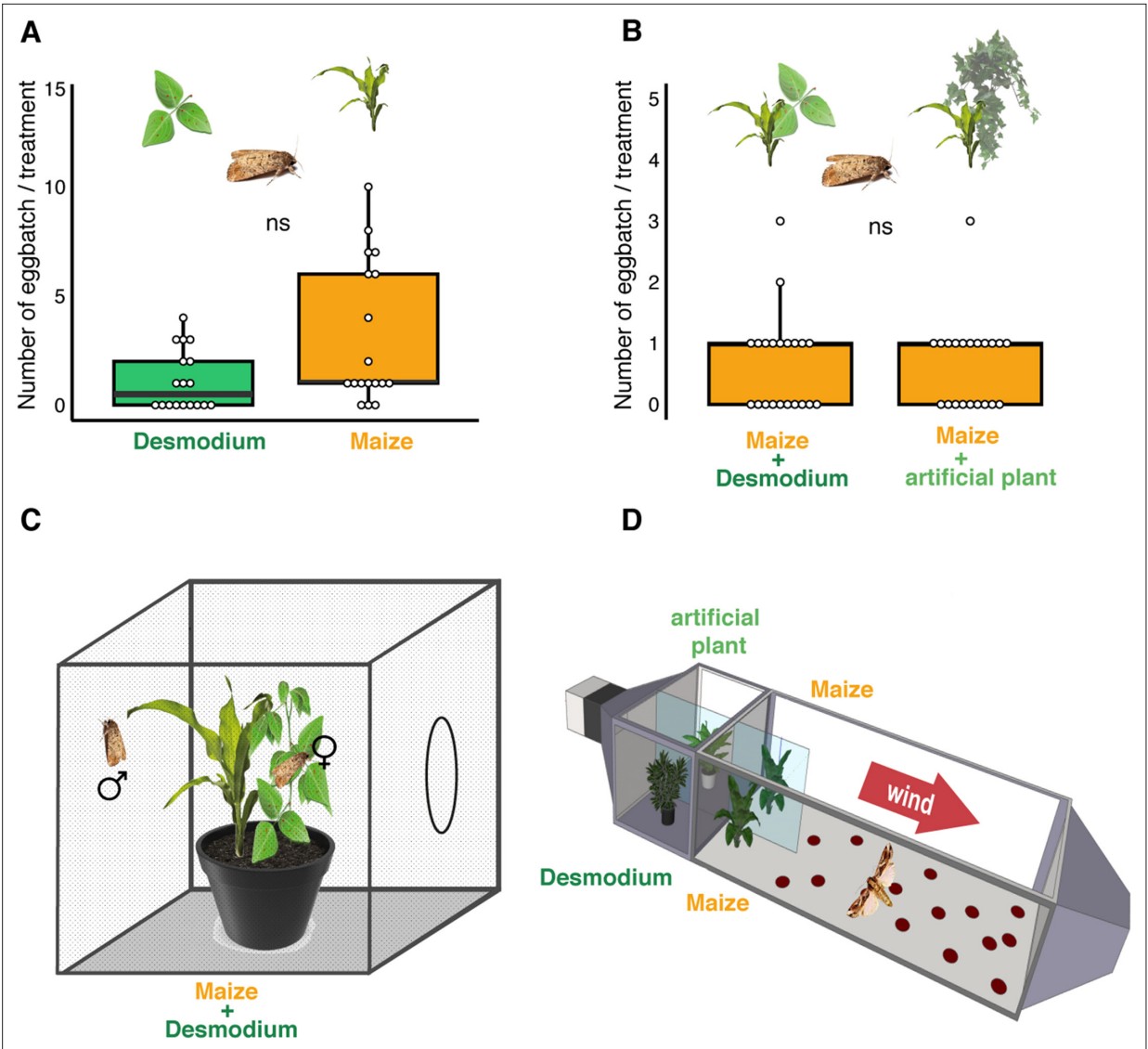

**Figure 3.** *D. intortum* does not repel ovipositing *S. frugiperda*. (**A**) The number of egg batches laid on *D. intortum* or *Z. mays* plants (*n* = 25, Wilcoxon signed rank exact test, p = 0.075, ratio of egg batches on other surfaces = 23%) in cage oviposition experiments (setup depicted in C). (**B**) Number of egg batches on *Z. mays* plants in a background of either *D. intortum* plant or a plastic plant mimic did not differ in wind tunnel oviposition assays (*n* = 21, Wilcoxon signed rank exact test, p = 0.825, ratio of egg batches on walls = 27%). (**C**) Oviposition experiments were conducted in netted cages and (**D**) and a modified wind tunnel setup (*Figure 3—figure supplement 1*). Wind tunnel setup to study the oviposition repellency of *Desmodium intortum* volatiles (*Figure 3—figure supplement 2*). The number of eggs laid on *D. intortum* or *Z. mays* plants (*n* = 25, Wilcoxon signed rank exact test, p = 0.105) in cage oviposition experiments (setup depicted in C).

The online version of this article includes the following figure supplement(s) for figure 3:

**Figure supplement 1.** Wind tunnel setup to study the oviposition repellency of *Desmodium intortum* volatiles.

**Figure supplement 2.** The number of eggs laid on *D. intortum* or *Z. mays* plants (*n* = 25, Wilcoxon signed rank exact test, p = 0.105) in cage oviposition experiments (setup depicted in *Figure 2C*).

to elucidate the mechanism of preference, the data demonstrate that *Desmodium* is a palatable plant for dispersing larvae, yet does not support larval development.

In addition to stagnating development, we found that larvae moved slowly on *Desmodium* leaves and stems, and many were entirely immobilized, particularly visible at later larval instars. Closer scrutiny of *D. intortum* surfaces revealed a dense network of non-glandular, uniseriate, and uncinate trichomes, with densities and a distribution depending on the surface type (*Figure 6A, D, F*; *Figure 6—figure supplement 1*). The stems and main veins of the leaves were particularly densely populated with large

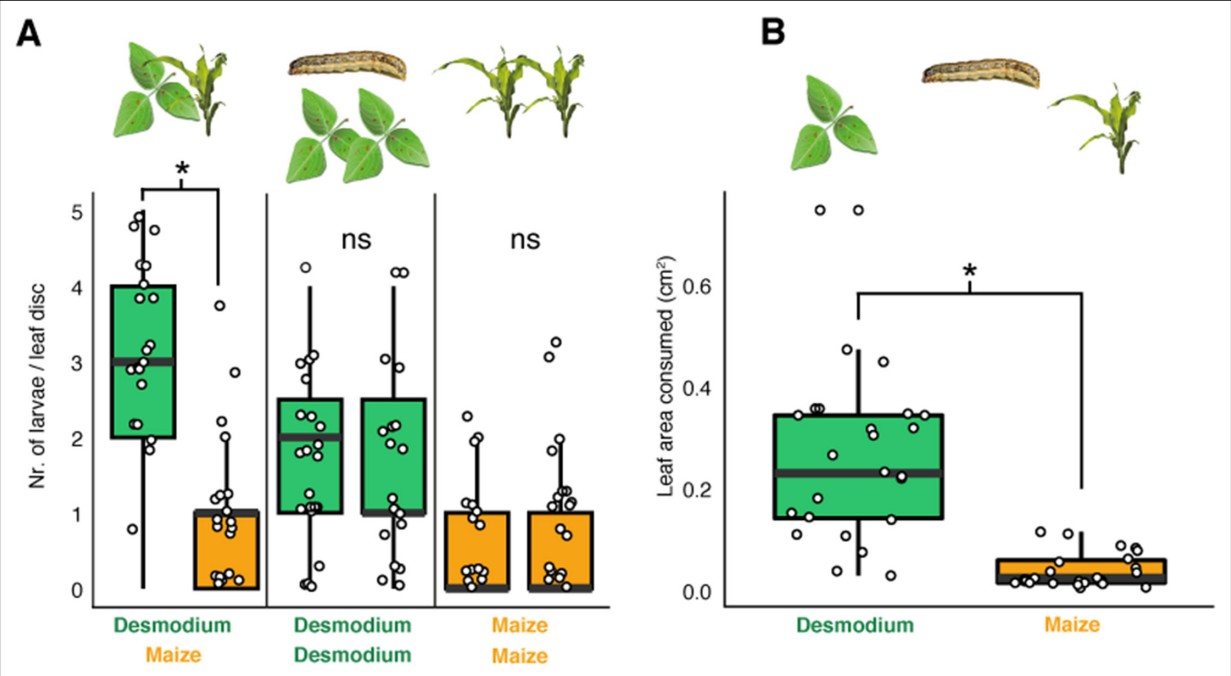

**Figure 4.** *D. intortum* is preferred by neonate *S. frugiperda* larvae. (**A**) First instar *S. frugiperda* larvae preferred *D. intortum* against *Z. mays* in two-choice leaf disc bioassays ($n = 25$, Wilcoxon signed rank exact test, $p = 2.73 \times 10^{-3}$). (**B**) First instar *S. frugiperda* larvae consumed more *D. intortum* than *Z. mays* (20 hr, two-choice leaf disc bioassays, $n = 25$, Wilcoxon signed rank exact test, $p = 3.338 \times 10^{-6}$, the ratio of non-settled larvae = 74.2%). The symbol * shows comparisions where p-values were lower than 0.05.

uncinate trichomes. First instar larvae were somewhat freely moving and grazing between trichomes of the stem (*Figure 6—figure supplement 1C*), but older larvae were seen impaled and immobilized by them (*Figure 6C, D*; *Figure 6—figure supplement 1D–F*). Occasionally, even ovipositing *S. frugiperda* were immobilized at their ovipositor on *D. intortum* (*Figure 6—figure supplement 1G*). Whereas trichomes were flexible at the base, they were fortified with silica toward the tip (*Figure 6F*), equipping the plant with an effective mechanism to obstruct, damage and immobilize herbivores. Also beneficial insects (*Figure 6—figure supplement 1I*) and even vertebrates can be trapped by *Desmodium* (*Coleman, 2016*). Incidentally, the presence of small uncinate trichomes on leaves could also have affected the movement of first instar larvae and thereby support the observed preference pattern observed earlier (*Figure 4*) and survival rate (*Figure 5*). Stellar non-glandular trichomes were shown to decrease feeding of *Manduca secta* larvae on several *Solanaceae* species (*Kariyat et al., 2018*), and uniserate non-glandular trichomes of bottle gourd (*Lagenaria siceraria*) affect the feeding and survival of *Trichoplusia ni* (*Kaur and Kariyat, 2023*). How the size, shape and density of these surface structures affect lepidopteran behavior in a species- and stage-specific manner needs to be addressed in future studies. Uncinate non-glandular trichomes are used by many other plant species (*Ballhorn et al., 2013*; *Gilbert, 1971*), and may serve multiple purposes including seed or fruit dispersal (*Xing et al., 2017*; *Sorensen, 1986*; *Freitas et al., 2014*).

We thus infer that in the field *Desmodium* trichomes affect fitness of lepidopteran larvae, both directly and indirectly. First, *Desmodium* entices larval feeding, but truncates development. Second, trichomes on *Desmodium* hinder movement, damage the cuticle and even entirely immobilize larvae on the plant, increasing developmental time, exposure to natural enemies and overall mortality (*Kaur and Kariyat, 2023*; *Kariyat et al., 2017*; *Kaur and Kariyat, 2020*). Third, ingestion of trichomes damages the intestinal lining and affects digestion, development and survival in closely related species (*Kaur and Kariyat, 2020*; *Acevedo et al., 2021*). Indeed, while first instar larvae fed around the large uncinate trichomes, larger larvae did ingest trichomes as evidenced by trichomes found in larval frass. Effectively, rather than functioning as a repellent intercrop, *Desmodium* appears to be a trap crop for larvae.

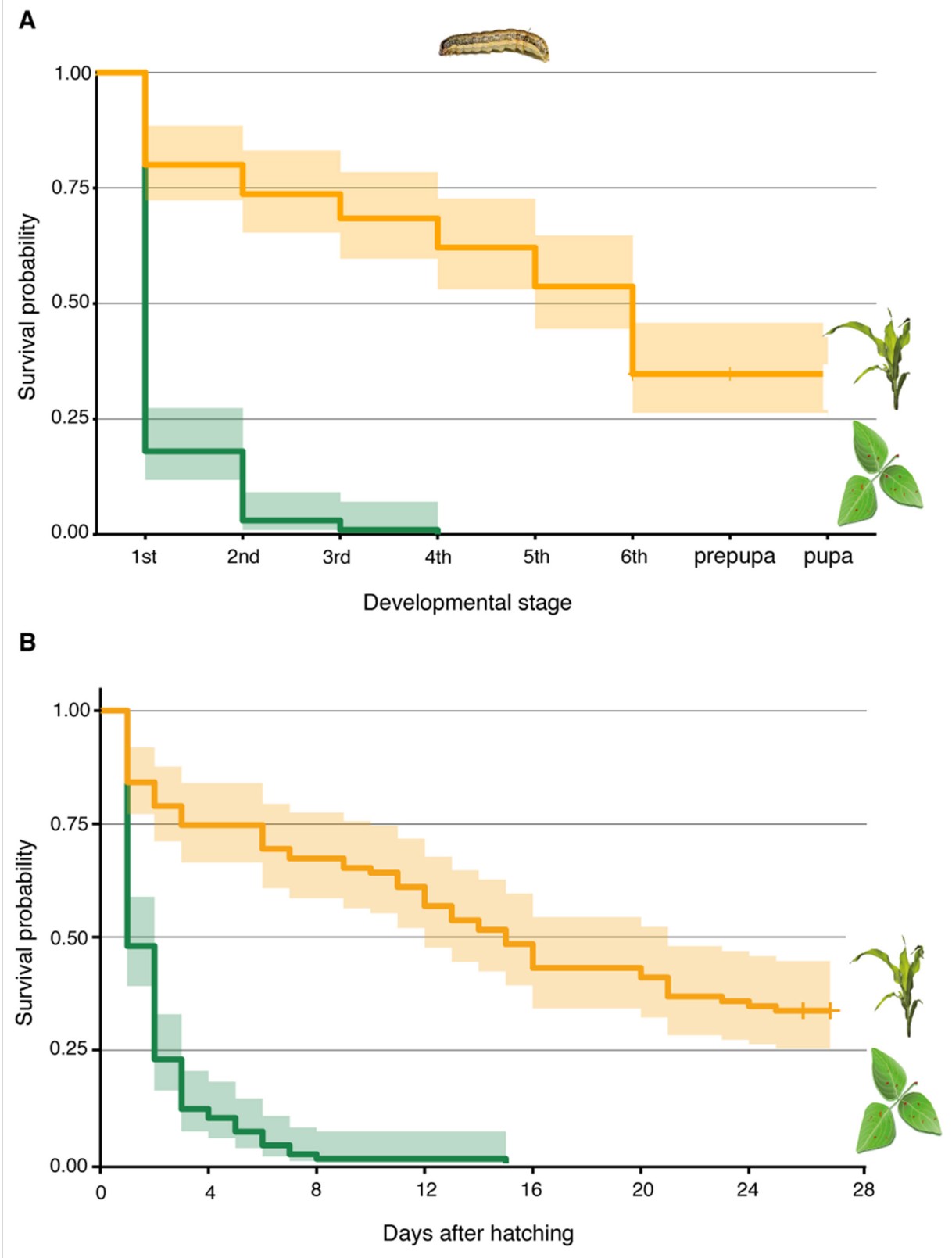

**Figure 5.** *D. intortum* is not a suitable host plant for *S. frugiperda*. (**A**) Survival probability of *S. frugiperda* on diets consisting of *D. intortum* (greenleaf *Desmodium*) was lower than on *Z. mays* in every developmental stage (n = 100, Kaplan–Meier survival analysis, p = 2.000 × 10$^{-16}$). (**B**) Larvae on *D. intortum* diet had significantly higher mortality throughout the experiment than larvae on *Z. mays* diet (n = 100, Kaplan–Meier survival analysis, p = 2.000 × 10$^{-16}$). The *D. intortum* diet resulted in a total mortality by the fourth instar larval stage. Envelope indicates the standard error.

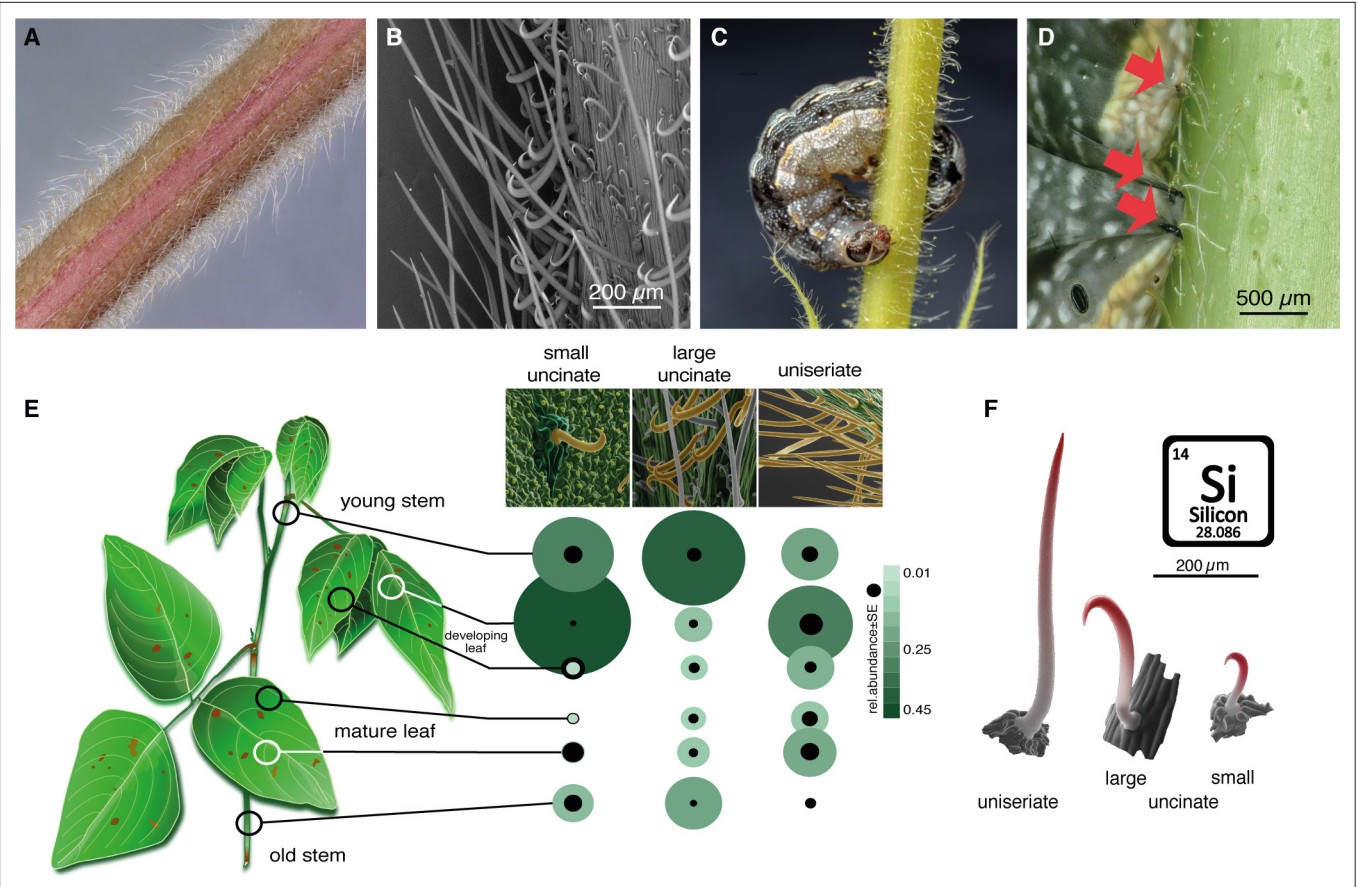

**Figure 6.** Non-glandular trichomes on *Desmodium intortum* act as a physical barrier for *Spodoptera* larvae. (**A**) Light microscopy image of a section of a young *D. intortum* stem densely covered with trichomes. (**B**) Scanning electron microscopy (SEM) image of a young *D. intortum* stem. Straight uniseriate hairs (up to 2 mm long) extended beyond the large (0.2–0.4 mm) and small (0.05–0.2 mm) hooked uncinate trichomes (scale bar: 200 µm). (**C**) A fifth instar *S. frugiperda* larva impaled and immobilized on a stem of *D. intortum* by both large and small uncinate trichomes. (**D**) Fourth instar *S. frugiperda* larva pierced by uncinate trichomes (red arrows). Trichomes either immobilized larvae or broke off from the basal cell with the tip remaining in the larval body causing severe wounds. (**E**) Distribution of non-glandular trichomes on different parts of the *D. intortum* plant. The relative abundance was calculated as the mean of trichome count divided by the sum of trichomes per trichome type across samples. Black circles indicate the standard error of relative trichome abundance (*n* = 5). (**F**) SEM images combining energy-dispersive X-ray spectroscopy (EDX) element topography images indicate relative surface silica (Si) distribution (red) of uniseriate, large, and small uncinate trichomes (*n* = 5) (*Figure 1*). *Spodoptera littoralis* larvae and adult *Spodoptera frugiperda* immobilized on *Desmodium intortum* and *Desmodium uncinatum* stems.

The online version of this article includes the following figure supplement(s) for figure 6:

**Figure supplement 1.** *Spodoptera littoralis* larvae and adult *Spodoptera frugiperda* immobilized on *Desmodium intortum* and *Desmodium uncinatum* stems.

We hypothesize that 'push' does not describe the mode of action of *Desmodium*. Instead, the plant exhibits properties reminiscent of a 'pull' crop, a 'trap crop'. Although superficially similar in mode of action to the 'pull' border crop Napier grass, *Desmodium* is distinctly different, as it is preferred by larvae, not by adults (*Khan et al., 1997b*; *Hassanali et al., 2008*). In addition, *Desmodium* forms a mechanical barrier to dispersing larvae. Further field studies need to detail how oviposition preference of different stemborer species, larval dispersal, development and survival on *Desmodium*, mechanical obstruction by *Desmodium*, and additional mechanisms such as parasitization and predation rates, interplay with crop phenology in suppressing various lepidopteran species across the cropping season. Knowing the exact mode of action is critical if we, for instance, wish to substitute the fodder crop *Desmodium* with a food crop to enhance food security, or design push–pull inspired, pest-suppressive conditions for other crops.

The observation that *Desmodium* does not emit detectable amounts of volatile terpenoids and does not repel *S. frugiperda*, contrasts with the large number of publications and the global attention

that maize–*Desmodium* push–pull technology has garnered over more than two decades. Indeed, the idea of the 'push' crop *Desmodium* repelling moths is found in numerous papers since its first mention around the year 2000. However, close scrutiny of the literature revealed a limited amount of primary data, except for a recent paper discussed below (*Sobhy et al., 2022*). The data presented here suggest that the mechanism of push–pull requires detailed studies. This is important as, although push–pull clearly suppresses lepidopteran pests (*Cook et al., 2007*; *Khan et al., 2010*; *Hassanali et al., 2008*; *Khan et al., 1997a*; *Khan et al., 2000*; *Khan et al., 2016*; *Midega et al., 2018*; *Feldmann et al., 2019*; *Mutyambai et al., 2019*), knowing the precise mechanism is essential to optimize the strategy to and troubleshoot it when it underperforms. How the mechanical defense described here impacts herbivore population, growth rate, the rate of parasitization and predation, depends on other biotic and abiotic factors and needs to be further studied as well. With the mechanism at hand, the strategy can also be further tailored to the needs of local smallholder farmers e.g. replacing *Desmodium* with food crops with similar properties (*Kariyat et al., 2018*; *Kaur and Kariyat, 2023*; *Ballhorn et al., 2013*; *Gilbert, 1971*; *Xing et al., 2017*; *Acevedo et al., 2021*; *Johnson, 1953*; *QulRing et al., 1992*), as well as rationally translating the concept to other cropping systems.

## Ideas and speculations

In our experiments, we exclusively detected volatile terpenoids when *D. intortum* was damaged by herbivores. In a recent study, aimed at testing the repellence of *Desmodium* to *S. frugiperda* under laboratory conditions, *D. intortum* plants may appear to emit volatile terpenoids constitutively (*Sobhy et al., 2022*). However, besides difference in odor collection methodologies (see below), the objective and experimental design of that study differed substantially from the current study, which makes comparison with the current study tenuous. Sohby et al. primarily aimed to assess the preference of *S. frugiperda* for maize alone or in combination with *Desmodium* without focusing on herbivore induction. At a first glance the volatile profile of intact *Desmodium* and maize in *Sobhy et al., 2022* would appear herbivore-induced, as these profiles compare well with our herbivore-induced *Desmodium* and maize. However, the absence of both positive and negative control plants (induced and non-induced *Desmodium* and other push-plants such as *M. minutiflora*) in *Sobhy et al., 2022*, makes a direct comparison difficult. The current study provided contrasts through the inclusion of these controls. *Desmodium*, under a large range of conditions in the laboratory and the field, did not release detectable amounts of induced volatile terpenoids, and comparatively little when induced. While the antenna of *S. frugiperda* females (*Sobhy et al., 2022*) can detect volatile terpenoids, in our study these appeared only to be released in detectable, small amounts upon herbivory of *Desmodium* plants.

Besides differences in the hypotheses and design of the studies, a methodological factor, the volatile collection methods, is also worth considering. Static (this study) versus dynamic headspace sampling (*Sobhy et al., 2022*) in combination with other factors such as the adsorbent used, impacts volatiles collected in many ways, and in turn its interpretation (*Tholl, 2020*; *Tholl et al., 2006*; *Raguso and Pellmyr, 1998*; *Ouyang and Pawliszyn, 2008*; *Prosen and Zupančič-Kralj, 1999*). Solid-phase microextraction (SPME) used here allows for time and cost-efficient collection of large numbers of samples, but this is non-exhaustive and with limited possibilities of absolute quantification (*Figure 7*). Compared to SPME, using adsorbent filled volatile traps in a dynamic headspace is an exhaustive volatile collection method (*Figure 7*) and its sensitivity and quantifiability is better compared to SPME (*Tholl, 2020*; *Ouyang and Pawliszyn, 2008*). In addition, the vastly more tedious collection procedures of dynamic headspace sampling substantially reduce the number of biological replicates that can be handled (*Tholl, 2020*; *Prosen and Zupančič-Kralj, 1999*).

The limitations of these methods point out the importance of designing the extraction protocols carefully including relevant blank samples and using internal reference compounds. Furthermore, the choice of method needs to be anchored in the research questions and hypotheses, which influences the design of the experiments and the choice of relevant positive and negative controls to include.

Further studies should quantitatively evaluate emission of behaviorally active volatile terpenoids from maize, *Desmodium* and e.g. Napier grass under various realistic field (intercrop) conditions. These will be helpful to understand odor release against a background of volatile terpenoids in the cropping system, and how insects may navigate in these odor spaces. Other factors that deserve further study include for instance the root–root and root–microbiome mediated interactions, as well

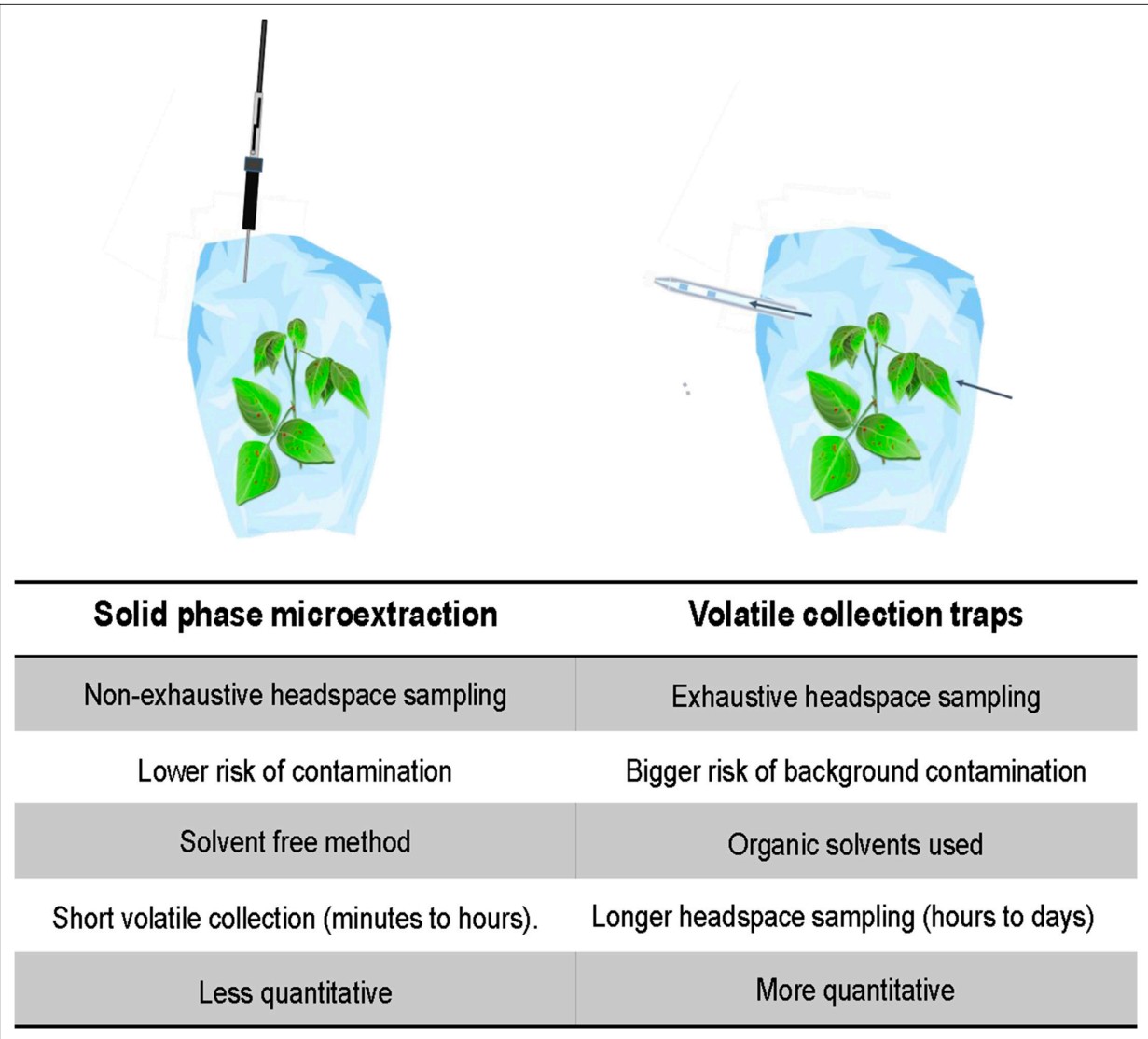

**Figure 7.** Comparison of volatile collection methods.

as possible plant–plant volatile communication. Recent studies have indicated their importance in shaping the above-ground chemical defenses (*Chen et al., 2019*; *Tao et al., 2017*; *Erb et al., 2015*). Since *Desmodium* and maize are planted in close proximity to each other, these root–root connections may further shape the plant–herbivore interactions under field conditions.

## Materials and methods
### Plants
Seeds of *D. intortum* (greenleaf *Desmodium*), and *D. uncinatum* (silverleaf *Desmodium*) were acquired from Simlaw seeds Co Ltd, Nairobi, Kenya. *M. minutiflora* seeds were obtained from the South African Sugarcane Research Institute (SASRI, Mount Edgecombe, South Africa). Maize seeds (*Zea mays* cv. Delprim) were provided by the laboratory of Professor Ted Turlings of the University of Neuchâtel, Switzerland. The cultivar is a European commercial hybrid and long-time standard, whose volatile emission patterns have been thoroughly studied (*de Lange et al., 2016*).

*Desmodium* spp. seeds were sterilized using 3% NaOCl, rinsed in distilled water and germinated on wet filter paper, and transferred to seedling trays with live or autoclaved soil (121°C for 20 min). After 21 days the plants were transferred to 18 cm diameter pots containing live or autoclaved soil and

were grown for 8 weeks in a greenhouse (22–25°C, light cycle 16:8 hr, 65% relative humidity). Another set of plants were raised from cuttings of mature stem parts of *D. intortum* and rooted in distilled water. Rooted cuttings were then planted in pots containing autoclaved soil with different inoculants: 200 g soil of a Tanzanian push–pull field per each pot, autoclaved soil with 60 mg of *Rhizobium leguminosarum*, *Bradyrhizobium japonicum* mixture per each pot (equal portions of *Rhizobia* inoculant for *Phaseolus* beans, and soy beans from Samenfest GmbH., Freiburg, Germany) or autoclaved soil with 120 mg of mycorrhizal fungi inoculate for each pot (mixture of *Glomus intraradices*, *G. etunicatum*, *G. monosporum*, *G. deserticola*, *G. clarum*, *Paraglomus brasilianum*, *Gigaspora margarita*, *Rhizopogon villosulus*, *R. lutcolus*, *R. amylopogon*, *R. fulvigleba*, *Pisolithus tinctorius*, *Scleroderma cepa*, and *S. citrinum*, Wildroot Organic Inc, Texas). The microbial inoculants were premixed in autoclaved soil before plant inoculation. Plants from cuttings grown on autoclaved soil were used as control. *M. minutiflora* seeds were germinated in live soil in plastic trays, and the seedlings were transferred into pots with live soil after two sets of leaves appeared. Eight-week-old *M. minutiflora* and *Desmodium* spp. plants were used in the experiments. Maize seeds were planted directly into live or autoclaved soil in pots and maintained in the greenhouse for 6 weeks.

### Insect rearing

Fall armyworm *S. frugiperda* were obtained from Ted Turlings at University of Neuchâtel, Switzerland, and were raised on a soybean-based semi-artificial diet supplemented maize whorls. Third instar larvae were separated into groups of ten individuals in plastic boxes. Pupae were sexed and separated in rearing cages. Adults were provided with a 5% sucrose solution and 6-day-old adults were mated for 6 hr and used in oviposition experiments.

### Volatile collection

The plants grown in the greenhouse were enclosed in a 60 cm × 20 cm polyethylene (PET) oven bag (Toppits 'Bratschlauch', Melitta, Minden, Germany) above ground for 24 hr to saturate the headspace. Prior to sampling, 2 µl of 250 ng/µl nonane solution in hexane was injected onto a piece of filter paper into the oven bag 40 min prior to sampling. SPME fibers (DVB/CAR/PDMS 50/30 µm, Supelco, Sigma-Aldrich, Bellefonte, PA, USA) were conditioned at 250°C in the split/splitless injector of the gas chromatography coupled mass spectrometry (GC–MS) in split mode for 10 min. The SPME fibers were exposed to the closed headspace for 30 min. The volatile emission of intact, mechanically damaged, and herbivore-damaged plants were sampled. *D. intortum* plants were mechanically damaged by cutting ten randomly selected leaflets in half, perpendicularly to the midrib. For herbivore treatment, eight fourth to fifth instar and 12 hr starved *S. frugiperda* larvae were put on the plants. In the first sets of experiments the feeding period lasted for 48 hr before volatile sampling.

A time series experiment of volatile terpenoid emission following herbivory was performed on *D. intortum* and *Z. mays* cv. Delprim plants grown on autoclaved soil inoculated with Tanzanian soil. Eight fourth instar larvae were put on each plant after 12 hr of starving and removed after 48 hr of feeding. The plants were sampled before herbivory and after 24 hr, 48 hr of herbivory. Larvae were removed from the plants after 48 hr and plants were resampled 72 hr and 1 week after the start of

**Table 1.** Volatile collection sites and environmental conditions.

| Sampling site | Practice | GPS coordinates | Relative humidity (%) | Temperature (°C) |
|---|---|---|---|---|
| Kitagasembe village, Gwitiriyo ward, Tarime district, Mara region, Tanzania | *D. intortum* monoculture, Maize with common beans as intercrop (*Phaseolus vulgaris*) | −1.3, 34.4792 | 70 | 20–22 |
| Gwitiriyo, Gwitiriyo ward, Tarime district, Mara region | *D. intortum* monoculture | −1.266661, 34.488133 | 68 | 20–22 |
| Kyoruba village, Pemba Ward, Tarime District | *D. intortum* monoculture | −1.318, 34.520 | 75 | 20–22 |
| Vi Agroforestry center, Lubango Ward, Musoma District, Mara region | Push–pull farming with *D. intortum* intercropping | −1.53054, 33.857955 | 81 | 20–22 |
| RUCID centre, Mityana district, Uganda | *D. intortum* monoculture and maize monoculture | 0.437941, 32.042500 | 68 | 26 |

the experiment. The volatile headspace was closed for 24 hr before each sampling and the SPME sampling procedure was the same as described above.

Field volatile samples of *D. intortum* (greenleaf *Desmodium*) and *Z. mays* were collected on farmer fields in Tarime and Musoma districts in Mara region, Tanzania, and Rural Community in Development (RUCID) center, in Mityana district, Uganda (*Table 1*). Healthy *D. intortum* plants, and maize plants with visible herbivore damage were selected and enclosed in 60 cm × 20 cm PET oven bags for 18 hr overnight. The use of reference compound and the SPME volatile sampling procedure are the same as described above. Closed empty oven bags with the injected reference compound were also sampled on each volatile sampling day and samples were compared to blank samples.

## Gas chromatography coupled mass spectrometry

A GC–MS (Agilent technologies, 7890B GC coupled with 5977 MSD) was used for SPME analysis. Fibers were inserted into a 250°C splitless injection port with The split valve closed for 1 min. The GC was equipped with a DB-WAX column (60 m × 250 μm × 0.25 μm). The carrier gas was helium and the total column flow was 1.883 ml/min. The temperature program of the oven started at 50°C and held for 1 min, then it was increased by 10°C/min to 220°C and then by 20°C/min to 250°C. The final temperature was held for 1 min. The mass spectrometer was used in electron ionization mode 70 eV and the detector scanned in the 29–400 *m/z* range. Samples were also injected on a GC–MS equipped with an HP-5 column (Agilent technologies, 6890 GC coupled with 5975 MSD, column: 60 m × 250 μm × 0.25 μm), with similar inlet settings and carrier gas (helium). The oven program was as follows: the starting temperature was 40°C and it was held for 2 min and increased by 8°C/min to 230°C and held for 2 min. The solvent delay and mass spectrometry settings were the same as described above.

GC–MS results were analyzed using Agilent Mass Hunter B.08.00, the peaks were auto integrated with agile integrator and manual integration. Compounds were tentatively identified by matching their mass spectra with those found in MS Libraries (NIST11 and Wiley12). The identification was verified by comparing calculated Kovats retention indices (RI) to those published in the NIST WebBook database and PubChem database (*Table 2*), and comparisons with analytical standards, see list of synthetic compounds in *Table 3*.

## Oviposition choice experiments

We conducted two experiments to study the short-range/multimodal oviposition repellency and long-range/olfactory oviposition repellency of *D. intortum* for *S. frugiperda* females.

### Short-range/multimodal oviposition repellency experiments

In short-range/multimodal oviposition repellency experiments, maize seeds (*Z. mays* cv. Delprim) and *D. intortum* cuttings were co-planted in 18 cm diameter pots. The experiments were conducted 3–4 weeks after co-planting, when the biomass of each plant was roughly similar. Plants were placed in 30 × 30 × 30 cm net cages (Bugdorm, Megaview, Taiwan) in a climate chamber set to 25 ± 2°C, 65% ± 5% relative humidity and 16:8 hr L:D cycle. Six-day-old virgin *S. frugiperda*, one female and one male, were mated for 6 hr and females were allowed to oviposit for 48 hr. A cotton ball soaked in 5% sucrose solution was placed between the plants for adult feeding. The egg batches and the number of eggs per each batch were counted at the end of the second day on both plants and the cage surfaces.

### Long-range/olfactory oviposition repellency experiments

To score for spatial repellency of *D. intortum*, a modified wind tunnel (180 cm × 80 cm × 60 cm, 30 cm/s airflow) was used (*Figure 3—figure supplement 1*). At the furthest upwind part of the flight section of the tunnel, two 4- to 5-week-old maize plants (*Z. mays* cv. Delprim) were positioned at 60 cm from each other. Directly upwind from the maize plants and separated by a stainless steel gauze (100 mesh) an 8-week-old *D. intortum* or artificial plastic plant was placed. In both sections a 20-cm plexiglass sheet was placed in line with the airflow to separate the airflows (*Figure 3—figure supplement 1*). Two 6-day-old females and one 6-day-old male were climatized in a plastic cup for 3 hr. One hour prior to scotophase, the cup was opened and placed on a 20-cm high stand in the center of the wind tunnel, 120 cm downwind from the maize plants. A cotton ball soaked in 5% sucrose solution was placed in the chamber at the release point as a source of food. The position of the female and

**Table 2.** Identification of volatile components from field and laboratory volatile collection. Compounds were tentatively identified by matching their mass spectra with those found in MS Libraries (NIST11 and Wiley). The identification was verified by synthetic standards (Compound (standard)) and matching Kovats retention indices (Compound (RI)) found in literature for DB-WAX and HP-5 capillary columns.

| Chemical group | Compound (RI) | CAS | DB-WAX RI (calc) | DB-WAX RI (lib) |
|---|---|---|---|---|
| Reference compound | nonane | 111-84-2 | 900 | 900 |
| Acetate ester | isobutyl acetate | 110-19-0 | 995 | 1002 |
| Acetate ester | isoamyl acetate | 123-92-2 | 1114 | 1126 |
| Acetate ester | (Z)-3-hexenyl acetate | 3681-71-8 | 1316 | 1320 |
| Acetate ester | (Z)-2-hexenyl acetate | 56922-75-9 | 1332 | 1319 |
| Primary alcohol | 1-propanol | 71-23-8 | 1026 | 1035 |
| Primary alcohol | 1-butanol | 71-36-3 | 1122 | 1136 |
| Primary alcohol | (Z)-3-hexen-1-ol | 928-96-1 | 1382 | 1387 |
| Secondary alcohol | 3-hexanol | 623-37-0 | 1189 | 1189 |
| Secondary alcohol | 3-octanol | 589-98-0 | 1388 | 1396 |
| Dialkyl ketone | 3-hexanone | 589-38-8 | 1044 | 1042 |
| Dialkyl ketone | 2-heptanone | 110-43-0 | 1185 | 1184 |
| Dialkyl ketone | 3-octanone | 106-68-3 | 1262 | 1248 |
| Methyl ketone | 6-methyl-5-heptene-2-one | 110-93-0 | 1332 | 1341 |
| Aliphatic aldehyde | (E)-2-hexenal | 6728-26-3 | 1224 | 1218 |
| Aliphatic aldehyde | ctanal | 124-13-0 | 1280 | 1287 |
| Saturated fatty aldehyde | nonanal | 124-19-6 | 1398 | 1396 |
| Saturated fatty aldehyde | decanal | 112-31-2 | 1490 | 1498 |
| Monocarboxylic acid | acetic acid | 64-19-7 | 1410 | 1410 |
| Monocarboxylic acid | butanoic acid | 107-92-6 | 1614 | 1612 |
| Monocarboxylic acid | pivalic acid | 75-98-9 | 1566 | 1579 |
| Aaromatic hydrocarbon | toluene | 108-88-3 | 1027 | 1037 |
| Aromatic hydrocarbon | styrene | 100-42-5 | 1254 | 1254 |
| Benzaldehyde | benzaldehyde | 100-52-7 | 1537 | 1528 |
| Benzoate ester | methyl salicylate | 119-36-8 | 1783 | 1778 |
| Monoterpene | (E)-alloocimene | 14947-20-7 | 1404 | 1396 |
| Monoterpene | 3-carene | 13466-78-9 | 1148 | 1142 |
| Monoterpene | p-cymene | 99-87-6 | 1277 | 1265 |
| Monoterpene | (E)-4,8-dimethylnona-1,3,7-triene | 19945-61-0 | 1303 | 1302 |
| Monoterpene | limonene | 138-86-3 | 1211 | 1200 |
| Monoterpenoid | linalool | 78-70-6 | 1531 | 1540 |
| Monoterpene | β-myrcene | 123-35-3 | 1166 | 1165 |
| Monoterpene | (E)-β-ocimene | 3779-61-1 | 1251 | 1250 |
| Monoterpene | (Z)-β-ocimene | 3338-55-4 | 1233 | 1234 |
| Monoterpene | α-pinene | 80-56-8 | 1015 | 1015 |

*Table 2 continued on next page*

*Table 2 continued*

| Chemical group | Compound (RI) | CAS | DB-WAX RI (calc) | DB-WAX RI (lib) |
|---|---|---|---|---|
| Monoterpene | γ-terpinene | 99-85-4 | 1242 | 1250 |
| Monoterpene | β-pinene | 127-91-3 | 1161 | 1136 |
| Monoterpenoid | monoterp2 | - | 1164 | - |
| Monoterpenoid | monoterp3 | - | 1230 | - |
| Monoterpenoid | monoterp4 | - | 1252 | - |
| Monoterpenoid | (Z)-4,8-dimethylnona-1,3,7-triene | - | 1262 | 1274 |
| Monoterpenoid | monoterp6 | - | 1272 | - |
| Monoterpenoid | monoterp7 | - | 1277 | - |
| Monoterpenoid | monoterp8 | - | 1297 | - |
| Monoterpenoid | monoterp9 | - | 1305 | - |
| Monoterpenoid | monoterp10 | - | 1306 | - |
| Monoterpenoid | monoterp11 | - | 1308 | - |
| Monoterpenoid | monoterp12 | - | 1315 | - |
| Monoterpenoid | monoterp13 | - | 1371 | - |
| Monoterpenoid | monoterp14 | - | 1376 | - |
| Monoterpenoid | monoterp15 | - | 1399 | - |
| Monoterpenoid | monoterp16 | - | 1405 | - |
| Sesquiterpene | β-bisabolene | 495-61-4 | 1740 | 1727 |
| Sesquiterpene | β-caryophyllene | 87-44-5 | 1619 | 1604 |
| Sesquiterpene | (E)-β-farnesene | 18794-84-8 | 1668 | 1665 |
| Sesquiterpene | germacrene D | 23986-74-5 | 1744 | 1746 |
| Sesquiterpene | α-humulene | 6753-98-6 | 1699 | 1690 |
| Sesquiterpenoid | sesquiterp1 | - | 1493 | - |
| Sesquiterpenoid | sesquiterp2 | - | 1498 | - |
| Sesquiterpenoid | cyclosativene | 22469-52-9 | 1500 | 1490 |
| Sesquiterpenoid | α-copaene | 3856-25-5 | 1503 | 1497 |
| Sesquiterpenoid | ylangene | 14912-44-8 | 1523 | 1499 |
| Sesquiterpenoid | sesquiterp6 | - | 1533 | - |
| Sesquiterpenoid | sesquiterp7 ((Z)-α-bergamotene) | 18252-46-5 | 1547 | 1555 |
| Sesquiterpenoid | α-cedrene | 469-61-4 | 1552 | 1565 |
| Sesquiterpenoid | sesquiterp9 | - | 1561 | - |
| Sesquiterpenoid | sesquiterp10 | - | 1566 | - |
| Sesquiterpenoid | sesquiterp11 | - | 1588 | - |
| Sesquiterpenoid | α-santalene | 512-61-8 | 1591 | 1597 |
| Sesquiterpenoid | sesquiterp13 | - | 1594 | - |
| Sesquiterpenoid | sesquiterp14 | - | 1607 | - |
| Sesquiterpenoid | sesquiterp15 | - | 1648 | - |
| Sesquiterpenoid | (Z)-β-farnesene | 28973-97-9 | 1653 | 1652 |

*Table 2 continued on next page*

*Table 2 continued*

| Chemical group | Compound (RI) | CAS | DB-WAX RI (calc) | DB-WAX RI (lib) |
|---|---|---|---|---|
| Sesquiterpenoid | α-himachalene | 3853-83-6 | 1657 | 1649 |
| Sesquiterpenoid | sesquiterp18 | - | 1658 | - |
| Sesquiterpenoid | sesquiterp19 | - | 1665 | - |
| Sesquiterpenoid | sesquiterp20 | - | 1678 | - |
| Sesquiterpenoid | γ-curcumene | 28976-68-3 | 1704 | 1695 |
| Sesquiterpenoid | sesquiterp22 | - | 1705 | - |
| Sesquiterpenoid | sesquiterp23 | - | 1717 | - |
| Sesquiterpenoid | β-curcumene | 28976-67-2 | 1753 | 1744 |
| Sesquiterpenoid | sesquiterp25 | - | 1768 | - |
| *Sesquiterpenoid* | *α*-curcumene | 644-30-4 | 1784 | 1773 |
| Sesquiterpenoid | sesquiterp27 | - | - | - |
| Sesquiterpenoid | (*E,E*)-4,8,12-trimethyltrideca-1,3,7,11-tetraene | 62235-06-7 | 1809 | - |
| Sesquiterpenoid | cadine-1,4-diene | 16728-99-7 | 1816 | 1802 |
| Sesquiterpenoid | sesquiterp30 | - | 1972 | - |
| Sesquiterpenoid | sesquiterp31 | - | 2020 | - |
| Sesquiterpenoid | sesquiterp32 (β-caryophyllene oxide) | 1139-30-6 | 2023 | 2013 |
| Sesquiterpenoid | sesquiterp33 | - | 2036 | - |
| Sesquiterpenoid | sesquiterp34 | - | 2075 | - |
| Sesquiterpenoid | sesquiterp35 | - | 2139 | - |
| Sesquiterpenoid | sesquiterp36 | - | 2175 | - |
| Sesquiterpenoid | sesquiterp37 | - | 2269 | - |
| Unknown | butyl acetate | 123-86-4 | 1054 | 1059 |
| Unknown | comp2 | - | 1114 | - |
| Unknown | comp3 | - | 1163 | - |
| Unknown | comp4 | - | 1183 | - |
| Unknown | butyl butanoate | 109-21-7 | 1213 | 1221 |
| Unknown | 5-hepten-2-one | 6714-00-7 | 1253 | 1249 |
| Unknown | 2-heptanol | 543-49-7 | 1316 | 1315 |
| Unknown | trimethyl-cyclohexanone | 2408-37-9 | 1333 | 1335 |
| Unknown | anisole | 100-66-3 | 1344 | 1340 |
| Unknown | comp10 | - | 1380 | - |
| Unknown | comp11 | - | 1393 | - |
| Unknown | comp12 | - | 1399 | - |
| Unknown | comp13 | - | 1414 | - |
| Uknown | comp14 | - | 1442 | - |
| Unknown | comp15 | - | 1450 | - |
| Unknown | comp16 | - | 1569 | - |

**Table 3.** Suppliers and purity of synthetic standard compounds.
The synthetic standards were injected in DB-WAX and HP-5 columns to verify identification of headspace volatile components.

| Compound | CAS | Supplier | Purity |
|---|---|---|---|
| (E)-alloocimene | 673-84-7 | Sigma-Aldrich | 80% |
| β-bisabolene | 495-61-4 | preparative GC | 1 µg/µl |
| camphene | 79-92-5 | Sigma-Aldrich | 95% |
| 3-carene | 13466-78-9 | Sigma-Aldrich | 90% |
| β-caryophyllene | 87-44-5 | Sigma-Aldrich | ≥98.0% |
| β-caryophyllene oxide | 1139-30-6 | Sigma-Aldrich | 95% |
| α-cedrene | 11028-42-5 | Sigma-Aldrich | 95% |
| α-cubebene | 17699-14-8 | preparative GC | 1 µg/µl |
| m-cymene | 535-77-3 | Sigma-Aldrich | 99% |
| p-cymene | 99-87-6 | Sigma-Aldrich | 99% |
| α-farnesene | 502-61-4 | Sigma-Aldrich | 95% |
| (Z)-farnesol | 106-28-5 | Sigma-Aldrich | 95% |
| (Z)-β-farnesene | 28973-97-9 | Sigma-Aldrich | 99% |
| (E)-β-farnesene | 18794-84-8 | preparative GC | 1 µg/µl |
| germacrene D | 23986-74-5 | preparative GC | 1 µg/µl |
| isobutyl acetate | 110-19-0 | Sigma-Aldrich | ≥98.0% |
| isoamyl acetate | 123-92-2 | Sigma-Aldrich | 99% |
| 3-hexanone | 589-38-8 | Sigma-Aldrich | ≥97% |
| 1-hexanol | 111-27-3 | Sigma-Aldrich | 99% |
| 2-heptanone | 110-43-0 | Sigma-Aldrich | 99% |
| (E)-2-hexenal | 6728-26-3 | Fluka | 99% |
| (Z)-3-hexen-1-yl acetate | 928-96-1 | Sigma-Aldrich | 98% |
| β-humulene | 116-04-1 | preparative GC | 1 µg/µl |
| γ-humulene | 6753-98-6 | Sigma-Aldrich | 85% |
| limonene | 5989-27-5 | Sigma-Aldrich | 97% |
| linalool | 78-70-6 | Sigma-Aldrich | 97% |
| methyl jasmonate | 1211-29-6 | Sigma-Aldrich | 98% |
| methyl salicylate | 119-36-8 | Sigma-Aldrich | 99% |
| β-myrcene | 123-35-3 | Sigma-Aldrich | ≥90.0% |
| nonane | 111-84-2 | Fluka | 99% |
| nonanal | 124-19-6 | Sigma-Aldrich | 95% |
| β-(E)-ocimene | 13877-91-3 | Sigma-Aldrich | 70% |
| 1-octen-3-ol | 3391-86-4 | Fluka | 98% |
| 3-octanone | 106-68-3 | Sigma-Aldrich | 99% |

*Table 3 continued on next page*

Table 3 continued

| Compound | CAS | Supplier | Purity |
|---|---|---|---|
| 3-octanol | 589-98-0 | Sigma-Aldrich | 99% |
| α-pinene | 86-56-8 | Sigma-Aldrich | 97% |
| β-pinene | 18172-67-3 | Sigma-Aldrich | 99% |
| α-phellandrene | 99-83-2 | Sigma-Aldrich | 85% |
| α-terpinene | 99-86-5 | Sigma-Aldrich | 95% |
| γ-terpinene | 99-85-4 | Sigma-Aldrich | 97% |

the number of egg batches laid on each side of the chamber were recorded after scotophase, 12 hr following the start of the experiment.

## Larval choice experiments

We conducted two-choice feeding bioassays to determine the feeding preference of the first larval instar of *S. frugiperda*. We cut 8 mm diameter leaf discs from young leaves of 6- to 7-week-old maize plants and leaves of 10- to 12-week-old *D. intortum* plants. We put the leaf discs on wet filter paper discs at 60 mm apart from each other in 100 mm × 20 mm plastic Petri dishes. Ten 1-day-old *S. frugiperda* larvae were placed in each arena and the position of larvae was recorded after 1, 2, and 20 hr periods. After 20 hr feeding each leaf disk was photographed and the consumed surface area of each disk was determined by image analysis using ImageJ (version 1.53) (*Schneider et al., 2012*).

## Larval survival experiments

Larval survival on maize and *D. intortum* was scored in plastic Petri dishes (100 mm × 20 mm), which were lined with wet filter paper to increase humidity. Five first instar *S. frugiperda* larvae were moved to each arena on the day of egg-hatching and fed daily with ad libitum amounts of freshly cut *D. intortum* leaves or leaf blades of 4- to 5-week-old maize (*Z. mays* cv. Delprim). After reaching the fourth instar, the maize diet was supplemented with the ligule, leaf sheets and young stems of maize and the larvae were separated into individual plastic cups to prevent cannibalism. The growth of the larvae was monitored daily, and we determined the larval stage based on body coloration and the diameter of head capsules. We terminated the experiment after the insects pupated.

## Light microscopy of *Desmodium* spp.

Upper and mid stem branches as well as the leaves of healthy 8-week-old *D. intortum* plants were sampled for light microscopy. In addition, *S. littoralis* larvae that were immobilized on *D. uncinatum* and *D. intortum* stems and leaves were observed and photographed with a digital light microscope (Keyence VHX-5000, Keyence Corporation, Osaka, Japan) equipped with standard zoom lens (VH-Z20R magnification: ×20–200 and VH-Z100R magnification: ×100–1000). For detailed, high depth-of-field images, a photo stacking technique was used. Series of images were captured (50–100 depending on the size of the examined larvae) at different focus distances (step size, 20–40 µm). Subsequently, partially focused images were combined with Helicon Focus software (Helicon Soft Ltd, Kharkiv, Ukraine) into a high depth-of-field image.

## Scanning electron microscopy of *Desmodium* spp.

To get further insights in the structure of the *D. intortum* trichomes, scanning electron microscopy was performed on leaf and stem samples. Healthy leaves and stems were collected from 8-week-old and 1-year-old plants from the greenhouse, and scanned using a FEI Quanta 3D scanning electron microscope operating with a field emission gun electron source, equipped with SE (LVSED/ETD), BSE (vCD), and EDAX SDD EDS detectors. Low-vacuum mode (50–80 Pa specimen chamber pressure) was used in order to avoid sample charging, and to allow using plant material without sample fixation, dehydration, and sample coating. The accelerating voltage was 10–20 kV with 40–480 pA beam current.

Furthermore, the elemental composition of trichomes was studied using energy-dispersive X-ray spectroscopy, acquisition time: 50 s. Measurements were taken in four regions (base, lower and higher

middle, and tip) on the longer type of trichomes and from three regions in case of small uncinate trichomes.

## Statistical analysis

In case of each volatile sample the absolute peak areas were divided by the area of the internal standard peak to account for differences in volatile sampling efficiency. The volatile components were categorized into four compound groups: monoterpenoids, sesquiterpenoids, green leaf volatiles, and other volatiles. We calculated the total sum of peak areas for these volatile groups across samples for the laboratory volatile collections and field volatile collections by location. The volatile collections were further normalized across samples by dividing the absolute peak areas by the sum of the total area of the volatile group from the corresponding dataset.

The clustered heatmaps of volatile emission profiles were generated from $z$-scores calculated from the normalized volatile data using package pheatmap (*Kolde, 2019*). Jaccard dissimilarity indices were calculated from binary (presence/absence) standardized volatile data and non-metric multi-dimensional scaling (NMDS) was completed using the metaMDS function of package vegan in R (*Oksanen et al., 2013*). Permutational multivariate analysis of variance was completed on Jaccard dissimilarity indices using the adonis function of the vegan package. For assessing differences in the normalized volatile peak areas for ($E$)-DMNT and ($E$)-β-ocimene between groups Kruskal–Wallis tests and Wilcoxon rank sum tests were used from package stats with Benjamini and Hochberg p value correction (*R Development Core Team, 2013*).

We used Wilcoxon paired rank sum tests with a null hypothesis of random choice using package stats for two-choice oviposition experiments and larval choice experiments. As the statistical power of Wilcoxon paired rank sum tests are limited, we also fitted generalized linear mixed models (GLMM) by maximum likelihood with fixed factor for choice and random factor for replication on the two-choice oviposition data using package lme4 (*Bates et al., 2015*). We used the simulation-based test from package DHARMa (*Hartig, 2021*) to assess the goodness of fit for the complete model. The post hoc tests were completed with the emmeans package using Tukey's comparisons (*Lenth et al., 2018*).

Survival probabilities were calculated with Kaplan–Meier survival analysis (*Kaplan and Meier, 1958*) and the survival curves were compared using a log-rank test between diets in package survival (*Therneau and Lumley, 2015*). Survival curves were visualized using package survminer (*Kassambara et al., 2021*).

## Acknowledgements

We are grateful to Mr. Samuel Nyanzi at Rural Community in Development (RUCID) center, Mityana, and Dr. Fred Kabi of Makerere University for support with volatile collections in Uganda. We thank Ms. Eva Svensson from Lund University for help with autoclaving soil. We are thankful to Alex Berg for pictures of *Desmodium* spp. highlighting their trichomes as well as immobilized insect larvae and Ábel Szabó for his support in scanning electron microscopy. Special thanks to Prof. Ted Turlings' lab for providing maize (*Zea mays*) seeds and *Spodoptera frugiperda* colonies and Prof. Peter Anderson for *Spodoptera littoralis* used in some pictures. We thank Prof. Marie Bengtsson, SLU, Sweden for providing standards for identification purposes. The following funding agencies are acknowledged for making this project possible: Sida, Food Security Program (ABD, TD), Ekhagastifelsen (ALE), SLU Global (TD), the EU Erasmus Program (ES), and János Bolyai Research Scholarship of the Hungarian Academy of Sciences (BPM).

## Additional information

### Funding

No external funding was received for this work.

### Author contributions

Anna L Erdei, Conceptualization, Data curation, Formal analysis, Investigation, Methodology, Writing - original draft, Writing - review and editing; Aneth B David, Conceptualization, Investigation,

Visualization, Methodology, Writing - review and editing; Eleni C Savvidou, Advaith Chakravarthy, Investigation, Methodology; Vaida Džemedžionaitė, Validation, Investigation, Methodology; Béla P Molnár, Conceptualization, Formal analysis, Validation, Investigation, Visualization, Methodology, Writing - review and editing; Teun Dekker, Conceptualization, Data curation, Formal analysis, Supervision, Funding acquisition, Investigation, Methodology, Writing - original draft, Writing - review and editing

### Author ORCIDs
Anna L Erdei http://orcid.org/0000-0002-7819-1287
Béla P Molnár http://orcid.org/0000-0002-3192-0868
Teun Dekker http://orcid.org/0000-0001-5395-6602

### Decision letter and Author response
Decision letter https://doi.org/10.7554/eLife.88695.sa1
Author response https://doi.org/10.7554/eLife.88695.sa2

## Additional files

### Supplementary files
• MDAR checklist

### Data availability
Data associated with volatile analysis and behavioral bioassays are available on figshare at https://doi.org/10.6084/m9.figshare.19297730 and GC–MS raw data is available at https://doi.org/10.6084/m9.figshare.25592544.

The following datasets were generated:

| Author(s) | Year | Dataset title | Dataset URL | Database and Identifier |
|---|---|---|---|---|
| Erdei AL, David AB, Savvidou EC, Džemedžionaitė V, Chakravarthy A, Molnár BP, Dekker T | 2022 | The push-pull intercrop Desmodium does not repel, but intercepts and kills pest | https://doi.org/10.6084/m9.figshare.19297730 | figshare, 10.6084/m9.figshare.19297730 |
| Erdei AL, David AB, Savvidou EC, Džemedžionaitė V, Chakravarthy A, Molnár BP, Dekker T | 2024 | The push-pull intercrop Desmodium does not repel, but intercepts and kills pests - raw data | https://doi.org/10.6084/m9.figshare.25592544 | figshare, 10.6084/m9.figshare.25592544 |

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
