## [Editor Report]

This study addresses both commonly accepted and alternative hypotheses for the mechanism by which an intercrop supports pest control in push–pull agriculture, a promising and broadly recognized approach for sustainable intensification. The findings address a widely recognized gap in data on the mechanism underlying push–pull systems and thus can be important for work on pest control in agroecology as well as plant–herbivore interactions more generally. The support of claims is solid, combining observations of several different mechanistic aspects in an uncommonly broad range of relevant environments with clear reasoning regarding experimental design.

---

## [Decision Letter]

**Decision letter after peer review:**

Thank you for submitting your article "The push–pull intercrop *Desmodium* does not repel, but intercepts and kills pests" for consideration by *eLife*. Your article has been reviewed by 3 peer reviewers, one of whom is a member of our Board of Reviewing Editors, and the evaluation has been overseen by Jürgen Kleine–Vehn as the Senior Editor.

Essential revisions:

1. Despite the manuscript's emphasis on the lack of difference between maize and desmodium in oviposition, oviposition was not rigorously assessed. Considering that number of eggs varies by batch and egg batches can vary by host plants, claiming that oviposition behavior is unaffected by egg batch size is not solid evidence. Therefore, it is also necessary to provide information about the number of eggs or the size of the batch. Moreover, as biased oviposition may have occurred in a particular plant species, data comparing the ratio of plants that laid eggs and those that did not are required. Lastly, the authors should report proportion of eggs laid on plants in comparison to the cage.

2. In oviposition experiments, the difference in planting methods (co–planting and separated planting) between the cage–based bioassay and the wind tunnel–based bioassay impairs consistent interpretation of the results. Reviewer 2 pointed out that co–planting can change the secretion of volatile substances in plants, requiring further investigation of these substances. According to Reviewer 3, since co–planted plants do not reflect outdoor planting distance well, a re–experiment with separate plantings such as wind tunnels should be conducted. As both opinions are reasonable, we suggest that cage bioassays be conducted at least in separate plantings, since considering plant–plant communication goes beyond the core message of the current paper.

3. According to both reviewers, more clarification is needed regarding why and how caterpillars migrate on Desmodium rather than maize. There has been discussion that early instar caterpillars can move through hanging by thread, but it is unclear whether this can occur between maize and desmodium. To clarify this, we need to understand why and how the pest caterpillar actually prefers Desmodium (e.g., prefers a particular volatile? Do herbivores grow larger, but with lower survival rates?).

4. Explain why the total number of larvae are the lowest in Maize/Maize treatment.

*Reviewer #1 (Recommendations for the authors):*

– It appears that the authors generally responded adequately to the points I raised during the initial submission process.

– Nevertheless, a more comprehensive comparison would be with the study by Sohby et al., which should be compared with this study. A clear analysis of the methodological differences should be conducted in order to understand why Sohby et al. and this study produced different results.

– Furthermore, adverbs that are still difficult to scientifically verify are often used, which makes reading difficult (e.g. Astoundingly).

– Pest preference mechanisms remain unclear. More discussion is needed.

*Reviewer #2 (Recommendations for the authors):*

This MS reveals that plants that have long been said to push are not, in fact, doing so, but are trapping and killing pests, thereby reducing pest outbreaks. They showed the mechanism of trapping pest larvae in detail and analyzed a lot of volatiles from the plant and maize. The sample sizes are enough and static analysis are stable. However, additional experiments or additional explanations may be needed.

1) They showed the volatiles from Desmidum with different soils (including different microbes). That experiment is very important. But I think they should show also the volatiles both Desmidum and Maize when both plants sharing the soil, because it might be possible to interact both plants under the soil.

2) About the oviposition preference experiment, I think they should use the corn with Desmisum sharing soil also in the wind tunnel experiment, because of the same reason as above.

3) Please explain, why larvae prefer Desmidum, and how larvae find Desmisum.

4) I think they should mention about the possibility of plant communication between Maize and Desmoduim in the discussion.

5) Figure 4 A. I don't understand the reason the total number of larvae are the lowest in Maize/Maize among the set. Did many larvae not on the leaf disc when the Maize/Maize treatment?

6) I think they should mention the result of the soil experiment in the abstract even if there was no relation.

*Reviewer #3 (Recommendations for the authors):*

Along these lines, some statements still need rectification so that this manuscript is placed correctly in context. For example, this statement in the summary is no longer true and needs to be rectified: "Surprisingly, however, Desmodium does not constitutively release volatiles nor repels pests. Astoundingly, in spite of the massive research output on push–pull, no primary data on either volatile release or repellence of Desmodium has been published." It would now be appropriate to point out that the authors have not found abundant constitutive volatile release by Desmodium, and that primary data on Desmodium volatiles and behavioral effects on moths were published only recently despite the use of those expectations in models of push–pull (built on a system using different companion plants) for decades. Ideally, a relevant preprint could also be briefly addressed in the authors' discussion (https://www.researchsquare.com/article/rs–2535302/v1), although it is appropriate that the main effort of comparison is made with the peer–reviewed study from Sobhy et al.

Similarly, the authors should be cautious to use precise language.

– Exaggerated language is sometimes used to make a point, rather than emphasizing gaps and data using accepted terminology (astoundingly, deathtrap, etc.). This can be off–putting and distracts from the scientific argument, rather than building it up.

– I have the impression that "terpenoids" and "volatiles" are used interchangeably in this manuscript.

– I have the impression that "absence" is used when "infrequent observation", "low overall abundance" or similar is more accurate (volatile terpenoids from desmodium)

– The comment that active volatile collection runs the risk of accumulating more contaminants than passive sampling is not well supported, and in this context, it would be relevant to point out that the active collection system referred to uses a charcoal filter to scrub incoming air.

Some aspects of the bioassays are not standard and could be criticized, and so these need to be directly addressed and explained.

– Co–planting of maize with desmodium for the oviposition assay can be justified but seems to decrease the likelihood that moths differentiate between the plants. In push–pull fields, the two crops share soil but are commonly spaced so that they do not grow into each other when the maize is young (in part to avoid competition).

– Usually proportion of eggs laid on plants versus on walls etc. is reported – apologies if I overlooked this, but I didn't see it here.

– More critically, eggs and not only egg batches should be counted. Egg batches can vary greatly in size. This could make the difference between finding significantly less oviposition on desmodium versus maize, or not (Figure 3). It is the number of eggs (potential larvae) that is most relevant.

– To my knowledge, 12 hours is a long time to run a wind tunnel bioassay. Commonly, these are used to observe initial responses. What is the reason here, and how do the 12 h data differ from observations commonly made in wind tunnels (first choice, number of approaches within 5–10 min, and so on)? Why combine a very long wind tunnel assay with a relatively, although not uncommonly long (48 h) oviposition assay, rather than using the wind tunnel to observe early decisions?

– Similarly, many aspects of experimental design were not comparable between this study and Sobhy. Choice of plant comparisons is addressed, but not issues of animal and plant age, handling, and timing. It would be helpful to summarize these similarly as is done for the choice of plant comparisons. Generally, choices about moth age and handling and plant age and handling are critical for interpreting bioassay results and should be clearly explained (and backed by data and observations).

– Although the authors support an argument about how their results should compare to a more recent publication from Sobhy and colleagues, they do not address the electrophysiological results in the Sobhy study. Do those results affect conclusions about the different behaviors reported in this study, versus by Sobhy?

The authors should check for correct and appropriate scientific language and grammar, e.g.

– Remove some adverbs (make your point based on your evidence, not on repeating "surprisingly", "shockingly", etc.)

– " similar to those reported in earlier (10)" – reported on earlier? Reported from earlier studies?

– "developmental deathrap" – the preferred term in this literature is "trap crop"

– eight–week–old, five–week–old, etc. (not eight weeks old, five weeks old, etc.)

– "let to oviposit" ––> "allowed to oviposit"

[Editors' note: further revisions were suggested prior to acceptance, as described below.]

Thank you for resubmitting your work entitled "The push–pull intercrop *Desmodium* does not repel, but intercepts and kills pests" for further consideration by *eLife*. Your revised article has been evaluated by Jürgen Kleine–Vehn (Senior Editor) and a Reviewing Editor.

The manuscript has been improved but there are some remaining issues that need to be addressed, as outlined below:

Following careful review by reviewers, the authors should revise the manuscript in the following four areas:

1) The authors may not claim that Desmodium does not release terpenoids or other volatiles. Rather, they can state that in several experiments with an established method, they did not detect them or only in low amounts in comparison to maize plants.

2) They must acknowledge that there can be differences between the Desmodium measured in Kenya and what they grew in Sweden and measured in Tanzania. This is also very important for the larger discussion on push–pull. We are not working with a Desmodium inbred line and there are large environmental gradients.

3) It is almost certainly less sensitive to measure with an equilibrium technique than to quantitatively collect over time.

4) Plants in the field may be damaged and it is not unrealistic or misleading to measure volatiles from plants grown from the same stock and under similar conditions. The aim of the Sobhy study was detection in the study system, not to assess the causes of volatile emissions. So the authors should not imply that the resulting measurements are unrepresentative of push–pull because there was not strict control on the use of pristine plants.

---

## [Author Response]

Essential revisions:1. Despite the manuscript's emphasis on the lack of difference between maize and desmodium in oviposition, oviposition was not rigorously assessed. Considering that number of eggs varies by batch and egg batches can vary by host plants, claiming that oviposition behavior is unaffected by egg batch size is not solid evidence. Therefore, it is also necessary to provide information about the number of eggs or the size of the batch. Moreover, as biased oviposition may have occurred in a particular plant species, data comparing the ratio of plants that laid eggs and those that did not are required. Lastly, the authors should report proportion of eggs laid on plants in comparison to the cage.

We have included information on the number of oviposited eggs in the cage experiments and specified the ratio of eggs and egg batches laid on the cage or wind tunnel walls. We aimed at studying spatial repellency, and we considered the number of egg batches a reasonable indicator of that, as the decision to oviposit on a plant is known to be affected by host quality, whereas the relationship between clutch size and host quality is more stochastic/tenuous (e.g. Silva et al. 2017, scientia agricola) and frequently poorly documented and therefore difficult to link to literature. In our case the number of oviposited eggs hold a par with the pattern observed in number of egg batches in our cage experiments. We added a supplementary figure to show that. We hope that the above satisfies the reviewer's comments.

2. In oviposition experiments, the difference in planting methods (co–planting and separated planting) between the cage–based bioassay and the wind tunnel–based bioassay impairs consistent interpretation of the results. Reviewer 2 pointed out that co–planting can change the secretion of volatile substances in plants, requiring further investigation of these substances. According to Reviewer 3, since co–planted plants do not reflect outdoor planting distance well, a re–experiment with separate plantings such as wind tunnels should be conducted. As both opinions are reasonable, we suggest that cage bioassays be conducted at least in separate plantings, since considering plant–plant communication goes beyond the core message of the current paper.

We conducted earlier preliminary experiments that did not show any influence of co-planting on volatile emission, similar to the absence of an effect of soil inoculation on volatile emission by maize or Desmodium. When we were setting up the experiments we reasoned that co-planting would be both more practical, and more reflective of the natural situation in the field.

The recommended distance between Desmodium and maize in the field is 37.5 cm, whereas in our experiments the distance was 16-17 cm. In our experiments maize and desmodium plants were grown together for only three weeks, compared to a field situation, where the Desmodium plant is growing together with maize for the whole season. In addition, farmers tend to keep Desmodium on the field for extended periods (several years, as it is seen as an investment). These plants have larger root networks and interactions with maize roots are presumably more extensive than what we had in our pots. Thus, besides not having seen such interactions, we believe it is safe to assume that the interaction between the rhizospheres is accordingly smaller and root-root interactions are less in this laboratory setting compared to intercropping fields. However, root-root interactions deserve indeed further detailed study where the focus is on how root-root interactions shape the volatilome. We noted an error in our MATERIALS AND METHODS: the pot sizes were not 12 cm but 18 cm for the co-planted pots. The 12 cm pots were used for single plants, this misinformation was corrected accordingly.

In summary, the question of root-root interactions is an interesting factor. However we feel that it is beyond the scope of the present paper. In addition, these experiments involving the preparations, such as planting of Desmodium, rearing, and conducting behavioural experiments and analyses would take well over half a year. We unfortunately do not have these personnel resources currently (Aneth David graduated and left, while Anna Erdei received another position), and hope that the above extra data and reasoning will suffice at present statements.

We have corrected the Materials and methods section. We also highlighted toward the end of the discussion pointing out that below ground interactions, including root-root interactions, are still understudied and would deserve further research.

3. According to both reviewers, more clarification is needed regarding why and how caterpillars migrate on Desmodium rather than maize. There has been discussion that early instar caterpillars can move through hanging by thread, but it is unclear whether this can occur between maize and desmodium. To clarify this, we need to understand why and how the pest caterpillar actually prefers Desmodium (e.g., prefers a particular volatile? Do herbivores grow larger, but with lower survival rates?).

When it comes to ‘migration’ we would rather use the term ‘dispersal’, which is more of a stochastic, non-directional process. It is well known that larvae of many lepidopteran species disperse. The ecological significance is that this reduces sibling competition, as egg batches, containing tens to hundreds of eggs on a single plant, are far too large to support the development of all hatching larvae. While first instar larvae also disperse on a plant through walking, dispersal between plants by first instar larvae occurs through silk threads spun on emergence. The silk thread together with the just emerged larvae is picked up and dispersed with wind. First instar S. frugiperda larvae have very well developed silk glands, which are particularly used for this purpose (Marti et Rogers, Annals of the Entomological Society of America, 1988). But instars of many other lepidopteran larvae that infest maize have also been observed to do this, including other Spodoptera species, Chilo partellus and Buseola fusca (Sokame et al., Entomologia Experimentalis et Applicata 2020).

Given that Desmodium is a continuous intercrop with much vegetative surface compared to maize particularly earlier in the growth season, there is a big chance of larvae ending up on Desmodium instead of maize during this non-directional dispersal. Since L1 larvae appear to prefer Desmodium, but do not develop on it, Desmodium acts as a dead-end host. We do not know the underlying causes of this phenomenon (eg secondary plant metabolites, low nutritional value etc.), and leave this interesting observation as a lead for further studies. Such studies can be highly relevant when searching for alternative intercrops to Desmodium (which is not much favoured by farmers).

Dispersal also occurs at later stages in larval life, but this is not with wind as larvae are too heavy to be carried by silk threads. The extent to which older larvae disperse over plant and ground surfaces is though less clear and arguably more difficult and dangerous and therefore an evolutionarily less stable strategy. Also in these cases, the chances of larvae ending up on Desmodium on the interrow is stochastically very high compared to finding single stalks of maize. Here again, larvae are ill fated when ending up on Desmodium, with an additional mechanism coming into play: hindrance by spines and hooks. Survival rate is likely very low, as indicated by our study.

Our paper thus covers experimentally more of the how than the why (evolutionary, ecological as well as nutritional background of dispersal and preference). This fits with the scope and hypotheses of the work. However, the points of ‘why’ raised by the reviewers are interesting and hopefully will give rise to further studies.

When it comes to larval choice: we do not know what factors culminated into a ‘preference’ for Desmodium. For instance, a likely explanation could be that the leaf surface is much softer than maize, with silicone embedded in the hairs rather than in the surface. Especially for L1 larvae the softness of the surface is a very important factor in the ability to feed and survive. Additional factors that could come into play include for instance taste compounds or volatiles that are released upon feeding, the perceived nutritional value such as protein content of leaves (Chen, Ruberson and Olson, Entomol Exp Appl. 2008, https://doi.org/10.1111/j.1570-7458.2007.00662.x). Possibly, the small uncinate hairs also contribute by hindering even small larvae in their movement. While the underlying mechanism of ‘choice’ thus remains unresolved at this point, the ecological significance is that larvae dispersing onto Desmodium are likely to stay and feed on this intercrop, which lowers pest pressure on maize. We hope that the above explanation, along with the extensive revision of this manuscript suffices.

4. Explain why the total number of larvae are the lowest in Maize/Maize treatment.

In the L1 instar phase dispersing larvae are very motile. It is possible that on the Desmodium leaves larval movement is hindered or possibly even somewhat arrested by the small uncinate trichomes. This is an effect that is independent of the choice setting and presence of maize.

Earlier studies on non-glandular trichomes have shown that they decrease feeding intensity. These experiments investigated trichomes with different morphologies for example stellar and uniseriate trichome types that do not cause arrest as uncinate (hooked) trichomes. Possibly, the increased feeding is a transient phenomenon and trichomes might affect larvae in an instar-dependent manner.

Reviewer #1 (Recommendations for the authors):– Nevertheless, a more comprehensive comparison would be with the study by Sohby et al., which should be compared with this study. A clear analysis of the methodological differences should be conducted in order to understand why Sohby et al. and this study produced different results.

We have substantially reformatted the parts of the manuscript including the comparison to the Sobhy et al. paper and improved the discussion by discussing the results of the electrophysiological recordings in that paper.

Note that after rereading our manuscript and having received comments from some extra pairs of eyes, we felt that the flow was much disrupted by the extended comparison with the Sohby paper in the middle of the Results section. We therefore fused that paragraph with the methodological comments at the end of the paper in the section 'ideas and speculations'.

– Furthermore, adverbs that are still difficult to scientifically verify are often used, which makes reading difficult (e.g. Astoundingly).

We fully agree. We went through the manuscript once more and trust that the revision has taken care of phrases and words of a similar kind.

– Pest preference mechanisms remain unclear. More discussion is needed.

We have substantially rewritten the results and discussion and included more insights on the possible mechanisms that can be involved in the observed preference and feeding pattern and also included comparison to further literature. See also answers above. Note that the ‘preference’ can be due to a large number of underlying factors (involving multiple sensory modalities – olfaction, taste, mechanical-, nutritional) acting alone or in combination and resulting in what is loosely defined as ‘preference’. We believe that the mechanism of larval preference is out of scope for this manuscript, but the reviewer is right, that this warrants further in-depth studies aimed at a better understanding of behaviour. We have attempted to point this out in the revision.

Reviewer #2 (Recommendations for the authors):This manuscript reveals that plants that have long been said to push are not, in fact, doing so, but are trapping and killing pests, thereby reducing pest outbreaks. They showed the mechanism of trapping pest larvae in detail and analyzed a lot of volatiles from the plant and maize. The sample sizes are enough and static analysis are stable. However, additional experiments or additional explanations may be needed.1) They showed the volatiles from Desmidum with different soils (including different microbes). That experiment is very important. But I think they should show also the volatiles both Desmidum and Maize when both plants sharing the soil, because it might be possible to interact both plants under the soil.

We have extensively addressed the concern of below ground root-root interactions above. We hope that the response and amendments in the manuscript address the point raised.

2) About the oviposition preference experiment, I think they should use the corn with Desmisum sharing soil also in the wind tunnel experiment, because of the same reason as above.

We appreciate this comment, and in fact the below ground interactions were the start of our studies resulting in the current manuscript. We indeed tested volatile emissions from maize and Desmodium under different regimes (soils) with and without co-planting. There was no difference in the resulting headspace. In subsequent cage experiments, we tested co-planted maize and Desmodium, which tests if co-planting changes oviposition rate and preference. Based on these finding, we would not expect differences whether co-planted or not. We went on to further test preference in the wind tunnel. These tests addressed another question, that of whether oviposition on maize is influenced by the volatile background of Desmodium. Since Desmodium has a similar volatile profile whether co-planted or not, we would not expect any difference if we instead would have used co-planted Desmodium and maize. In addition, more practically, if co-planted plants would have been used, similarly to the cage experiments, it would have been impossible to separate olfaction from taste and other tactile stimuli in the ovipositing moth. Regardless of this practicality, we agree that there is room for further experiments, yet, the data presented give no indication to expect dramatic changes in preference and do not warrant more of these experiments. We do, however, indicate in the final part of the manuscript that further detailed experiments on microbe-root and root-root interactions are warranted, as literature evidence over the last decades indicates that such roles, particularly of microbiome-root interactions, may further shape plant insect interactions.

3) Please explain, why larvae prefer Desmidum, and how larvae find Desmisum.

The first instar S. frugiperda larvae are very motile and can disperse using silk threads over larger distances (several meters). This dispersion is a stochastic process that is not directed by the larval senses. In the field setting where maize and Desmodium is intercropped, the dispersion onto the Desmodium row from maize is highly likely, particularly since Desmodium is a continuous perennial interrow crop.

When it comes to feeding preference and choice in the behavioral setup, this can be driven by multiple senses such as olfaction (volatiles released particularly upon feeding), gustation (gustatory stimuli that suggest a certain quality of the plant), mechanical stimuli (the toughness of the leaf surface, which is particularly important in early larval stages), and vision (reflection of the leaf surface). Mechanical hindrance may also interplay here, as indicated above. Although L1 larvae appeared to move freely over the Desmodium leaf surface, the small uncinate trichomes could possibly slowed or arrested larvae to some extent and induce prolonged feeding. Whether the observed ‘preference’ is due to choice or due to other factors is not known, but the question is ecologically not so relevant, as L1 larvae do not have the chance of ‘choosing’ between the two host plants, as they are either located on maize or on Desmodium. What is important here is that L1 larvae are not at all inhibited to eat from Desmodium and not pressed to disperse further because of unpalatability of Desmodium. Later larval instars are much more hindered by trichomes and can be entirely arrested and killed by them. Here the question of whether they will eat from Desmodium becomes somewhat less relevant as other mechanisms exist that truncate larval development of later stages.

In other studies on non-glandular trichomes larvae appeared to decrease feeding intensity (Kariyat et el. 2017). However, these experiments investigated trichomes with different morphologies for example stellar and uniseriate trichome types that do not hinder movement as much as uncinate (hooked) trichomes. We hypothesise that the increased feeding is a transient phenomenon particularly important for first larval instars. In addition, trichomes might affect larvae in an instar-dependent manner. Further research is needed to understand the mechanisms of herbivore-suppression by uncinate trichomes and their physiological effects on larval development and population dynamics in the field. This we highlight in our discussion.

4) I think they should mention about the possibility of plant communication between Maize and Desmoduim in the discussion.

Thank you for the suggestion. Plant-plant communication through volatiles is a relevant aspect of the intercropping system that is as of yet understudied. We added a line that such communication may further shape the outcome of Desmodium-maize intercropping.

5) Figure 4 A. I don't understand the reason the total number of larvae are the lowest in Maize/Maize among the set. Did many larvae not on the leaf disc when the Maize/Maize treatment?

There is a stochastic effect caused by a combination of the high motility of L1 larvae, combined with a lower ‘preference’ for maize. The leaf surface of maize, even of young maize leaves used here, is tough, particularly for small larvae. As a result larvae tended to move more and therefore at any moment in time fewer larvae resided on the leaves (which is what we scored) than in Desmodium (which elicited feeding more efficiently), in spite of having the same number of larvae in the Petri dish. In addition, it may be that unicate trichomes lowered motility of larvae on Desmodium and thus increased the time spent on the leaves.

6) I think they should mention the result of the soil experiment in the abstract even if there was no relation.

Thank you for the suggestion. We included the description of this experiment in the abstract.

Reviewer #3 (Recommendations for the authors):Along these lines, some statements still need rectification so that this manuscript is placed correctly in context. For example, this statement in the summary is no longer true and needs to be rectified: "Surprisingly, however, Desmodium does not constitutively release volatiles nor repels pests. Astoundingly, in spite of the massive research output on push–pull, no primary data on either volatile release or repellence of Desmodium has been published." It would now be appropriate to point out that the authors have not found abundant constitutive volatile release by Desmodium, and that primary data on Desmodium volatiles and behavioral effects on moths were published only recently despite the use of those expectations in models of push–pull (built on a system using different companion plants) for decades. Ideally, a relevant preprint could also be briefly addressed in the authors' discussion (https://www.researchsquare.com/article/rs–2535302/v1), although it is appropriate that the main effort of comparison is made with the peer–reviewed study from Sobhy et al.Similarly, the authors should be cautious to use precise language.– Exaggerated language is sometimes used to make a point, rather than emphasizing gaps and data using accepted terminology (astoundingly, deathtrap, etc.). This can be off–putting and distracts from the scientific argument, rather than building it up.

We fully agree. We corrected the text accordingly.

– I have the impression that "terpenoids" and "volatiles" are used interchangeably in this manuscript.

Thank you for the suggestion, we have altered the wording to ‘volatile terpenoids’ to clarify the distinction.

– I have the impression that "absence" is used when "infrequent observation", "low overall abundance" or similar is more accurate (volatile terpenoids from desmodium)

Thank you for the suggestion, we have changed the text.

– The comment that active volatile collection runs the risk of accumulating more contaminants than passive sampling is not well supported, and in this context, it would be relevant to point out that the active collection system referred to uses a charcoal filter to scrub incoming air.

We have removed this statement from the description of volatile collection methods.

Some aspects of the bioassays are not standard and could be criticized, and so these need to be directly addressed and explained.– Co–planting of maize with desmodium for the oviposition assay can be justified but seems to decrease the likelihood that moths differentiate between the plants. In push–pull fields, the two crops share soil but are commonly spaced so that they do not grow into each other when the maize is young (in part to avoid competition).

Indeed, in the field, Desmodium is in the interrow and does not overlap with maize in the early growth stages. Later, maize overshadows Desmodium and contact is frequent. It is indeed possible that the close proximity may have resulted in ‘oviposition errors’ by females and thus lowered the discrimination (although also demonstrating that Desmodium perse is not ‘repellent’ in the strict sense). We indeed expect that in the field the preference for maize is more pronounced due to this. We included a sentence in the Results section to highlight that. The wind tunnel studies complement the cage experiments by providing only a Desmodium odor background. No repellence was observed in these settings, complementing the cage experiments in that there was no evidence of spatial repellence by Desmodium.

– Usually proportion of eggs laid on plants versus on walls etc. is reported – apologies if I overlooked this, but I didn't see it here.

This data was only included in the public repository and we included it in this version of the manuscript.

– More critically, eggs and not only egg batches should be counted. Egg batches can vary greatly in size. This could make the difference between finding significantly less oviposition on desmodium versus maize, or not (Figure 3). It is the number of eggs (potential larvae) that is most relevant.

We have included the data on the number of eggs in the cage experiment in figure 4 supplement 2. The ratio of eggs and egg batches on the plants showed a very similar pattern. See also comments above.

– To my knowledge, 12 hours is a long time to run a wind tunnel bioassay. Commonly, these are used to observe initial responses. What is the reason here, and how do the 12 h data differ from observations commonly made in wind tunnels (first choice, number of approaches within 5–10 min, and so on)? Why combine a very long wind tunnel assay with a relatively, although not uncommonly long (48 h) oviposition assay, rather than using the wind tunnel to observe early decisions?

The wind tunnel was used to set up a choice bioassay and it was not used and interpreted as a wind tunnel assay to assess the initial responses of females. We aimed at designing an additional oviposition assay, where only the volatile emission of Desmodium plants could influence the oviposition on maize plants and the further visual, tactile or gustatory cues are excluded. We aimed at comparing the oviposition of females in the presence of Desmodium plants compared to the presence of Desmodium volatile emissions. We used this modified wind tunnel as we could fit the maize plants in the foreground and the Desmodium plants in the background, while simultaneously masking the additional visual cues.

Indeed, wind tunnel bioassays are often short of duration. In our experience, however, female oviposition choice can include a rather long response time. Animals can in some cases be more forced to make choices, e.g. through ageing, deprivation of mating, absence of stimuli following mating, lower female calorie intake (lower weight, easier flight) etc. but in our experiments we decided to provide a longer time window, such that females displayed their behavior more naturally and were not forced to display a certain behavior (which could perhaps skewing choices).

– Similarly, many aspects of experimental design were not comparable between this study and Sobhy. Choice of plant comparisons is addressed, but not issues of animal and plant age, handling, and timing. It would be helpful to summarize these similarly as is done for the choice of plant comparisons. Generally, choices about moth age and handling and plant age and handling are critical for interpreting bioassay results and should be clearly explained (and backed by data and observations).

Perhaps good to reiterate here that our paper was published online before Sohby et al.’s paper. We conducted our experimental work without knowing of each other’s study. Unfortunately, however, our study‘s peer review got delayed. It is though worth to evaluate if there is anything obvious in the experimental design that could lead to different results. Compared to Sohby et al’s Materials and methods, there were no obvious differences that could be suggested as possible cause to difference in results. We added this to our description in the Results section. We insert here a table comparing the two studies. We also highlighted the sections in the Materials and methods that detail the age of the plants and insects.

**Author response table 1. sa2table1:** 

experiment	specimen	Sobhy et al.	manuscript
volatile collection	Desmodium	no information	6- 8 weeks
wind tunnel oviposition	Desmodium		11 weeks
wind tunnel oviposition	Maize		4-5 weeks
cage oviposition	Desmodium	4-5 weeks	8 weeks
cage oviposition	Maize	2-3 weeks	3-4 weeks
wind/cage	S. frugiperda	2 days old mated	6 days old mated

A personal note: we try to avoid too detailed comparison as it comes with the risk of casting doubt over some of the results of our colleagues' work, while time will show (see also below). In our opinion, it serves no purpose when there are reasonable underlying causes of the difference, namely induction: the apparent absence of terpenoid volatiles can only be explained by that the amounts are below detection thresholds(our manuscript), while there are several possible scenarios (mites, aphids or other herbivores that can cause unnoticed damage, or even remaining egg batches in cages as is commonly found in Lepidoptera, perhaps below ground herbivory), that may have resulted in the observation of volatile terpenoids (Sohby et al.). Our study demonstrated in more than 300 samples from laboratory and field that volatile emission by uninduced healthy Desmodium is very low, generally below threshold, and Desmodium displayed a low inducibility upon S. frugiperda attack, while in contrast maize was much more inducible. We respect our colleagues' work, and would like to not detail the comparison too much. We trust this comes naturally in due time. We hope you agree.

– Although the authors support an argument about how their results should compare to a more recent publication from Sobhy and colleagues, they do not address the electrophysiological results in the Sobhy study. Do those results affect conclusions about the different behaviors reported in this study, versus by Sobhy?

The EAD data data only demonstrate sensitivity of the antennae to the compounds, which was already known from previous studies. How that translates into behavior is always a big question, as one cannot extract behavioral significance from sensory responses.

Some points of caution here: Sohby et al. did not include a positive control, making it hard to assess if the plants were indeed intact or what their level of damage was. It is interesting to note that the volatile profile of intact Desmodium plants showed a high variation, which was visible when comparing the GC-EAD recordings, even though they were described as coming from the same plant sample. In our study we have found that Desmodium plants do hardly emit detectable amounts of volatile terpenoids in the absence of herbivory, and low amounts following herbivory. This contrasted strongly with herbivore induced maize plants, both in the lab and field. The fact that the volatile profiles described by Sobhy et al. are similar to the volatile profiles of Desmodium plants under herbivory in our study, and the fact that the volatile profiles in that paper are highly variable between recordings hint that there might be an underlying effect of recent herbivory. Regardless, we do think that the GC-EAD of Sohby et al. highlight that, similar to maize, the emission of monoterpenoids by Desmodium plants during herbivory by S. frugiperda larvae can be detected by natural enemies and ovipositing females and this may have implications for pest suppression in field settings.

The authors should check for correct and appropriate scientific language and grammar, e.g.– Remove some adverbs (make your point based on your evidence, not on repeating "surprisingly", "shockingly", etc.)

We have corrected the text and removed the repetitive adverbs. We removed ‘astoundingly’ and ‘shockingly’ in our manuscript. We replaced ‘surprisingly’ by unexpectedly.

– " similar to those reported in earlier (10)" – reported on earlier? Reported from earlier studies?

We have changed to “similar to those reported in earlier studies for intact plants” as this part of the manuscript referred to results of earlier studies.

– "developmental deathrap" – the preferred term in this literature is "trap crop"

Changed accordingly.

– eight–week–old, five–week–old, etc. (not eight weeks old, five weeks old, etc.)

We have corrected these grammatical errors.

– "let to oviposit" ––> "allowed to oviposit"

Corrected

[Editors’ note: what follows is the authors’ response to the second round of review.]

The manuscript has been improved but there are some remaining issues that need to be addressed, as outlined below:Following careful review by reviewers, the authors should revise the manuscript in the following four areas:1) The authors may not claim that Desmodium does not release terpenoids or other volatiles. Rather, they can state that in several experiments with an established method, they did not detect them or only in low amounts in comparison to maize plants.

We agree. We intended to clarify this in the previous version, but apparently there were some points where this was not yet clear. We went thoroughly through the whole text to make sure that we did not find any detectable amounts of terpenoid volatiles. We indeed cannot say anything about what might have been below the analytical detection limits of our methods.

2) They must acknowledge that there can be differences between the Desmodium measured in Kenya and what they grew in Sweden and measured in Tanzania. This is also very important for the larger discussion on push–pull. We are not working with a Desmodium inbred line and there are large environmental gradients.

We agree that there is a rather diverse genetic background in Desmodium. This was already obvious in the phenotypic diversity displayed in the seedlings in greenhouse experiments. In the field other biotic and abiotic variations interplay in this variation. To account for variation, we sampled not only from greenhouse grown Desmodium (with seeds from Simlaw, Kenya), but also from the field from three geographic locations, in East African high and lowlands. We have tried to further accentuate the importance of variation in the text.

3) It is almost certainly less sensitive to measure with an equilibrium technique than to quantitatively collect over time.

We do agree and we addressed this point extensively in the speculations section, in response to a previous review at *ELife*. We changed the wording to accentuate this.

4) Plants in the field may be damaged and it is not unrealistic or misleading to measure volatiles from plants grown from the same stock and under similar conditions. The aim of the Sobhy study was detection in the study system, not to assess the causes of volatile emissions. So the authors should not imply that the resulting measurements are unrepresentative of push–pull because there was not strict control on the use of pristine plants.

We agree with this comment, but were actually puzzled as we did not find where in the manuscript we suggested that the data presented in Sohby et al. are not representative of field conditions. We certainly do not mean to say that. We merely point out that the direct comparison of our data with those of Sohby et al. is tenuous as (1) at onset the questions in both studies were very different and accordingly the experimental protocols, and (2) Sohby et al. did not include the controls that would be necessary to make the comparisons. However, we would be happy to receive a more concrete suggestion on where it is we should modify the text.

In this context it is also important to note that, since what is observed in the greenhouse may not be representative of what happens under field conditions, we measured volatile production by Desmodium in the field in three geographic locations in East Africa. These show low and variable amounts of volatiles by Desmodium, much lower than maize, and similar to herbivore-induced Desmodium in greenhouses. The fact that there is emission of some volatiles, albeit very low, is in line with Sohby et al. that Desmodium can be induced to release volatiles.

Finally, we went through the text once more and attenuated the text where we felt it would help the flow and reduce the contrast with Sohby et al.